# IMPLICIT BIAS IN MATRIX FACTORIZATION AND ITS EXPLICIT REALIZATION IN A NEW ARCHITECTURE

## ABSTRACT

Gradient descent for matrix factorization is known to exhibit an implicit bias toward approximately low-rank solutions. While existing theories often assume the boundedness of iterates, empirically the bias persists even with unbounded sequences. We thus hypothesize that implicit bias is driven by divergent dynamics markedly different from the convergent dynamics for data fitting. Using this perspective, we introduce a new factorization model: $X \approx UDV^\top$, where $U$ and $V$ are constrained within norm balls, while $D$ is a diagonal factor allowing the model to span the entire search space. Our experiments reveal that this model exhibits a strong implicit bias regardless of initialization and step size, yielding truly (rather than approximately) low-rank solutions. Furthermore, drawing parallels between matrix factorization and neural networks, we propose a novel neural network model featuring constrained layers and diagonal components. This model achieves strong performance across various regression and classification tasks while finding low-rank solutions, resulting in efficient and lightweight networks.

## 1    INTRODUCTION

The Burer–Monteiro (BM) factorization (Burer & Monteiro, 2003b) is a classical technique for obtaining low-rank solutions in optimization. It can be viewed as a simple neural network that uses a single layer of hidden neurons under linear activation. Indeed, say one has the factorization $X = UV^T$ where $U \in \mathbb{R}^{d \times r}$ and $V \in \mathbb{R}^{c \times r}$, then $U$ and $V$ can be thought of as the weights of the first and second layers, while $r$ represents the number of hidden neurons. However, despite the similarity suggested by this view, there is a clear distinction between BM factorization and neural networks about how the rank $r$ is chosen. In BM factorization, $r$ is typically chosen to be small, close to the rank of the desired solution. Neural networks, on the other hand, often succeed even in overparametrized settings where $r$ is large.

Recent findings of implicit regularization in matrix factorization narrows the gap between these two perspectives. For instance, in an effort to explain the empirical success of overparametrized neural networks, Gunasekar et al. (2017) demonstrate that gradient descent (with certain parameter selection) on BM factorization tends to converge toward approximately low-rank solutions even when $r = d$. Based on this observation, they conjecture that *"with small enough step sizes and initialization close enough to the origin, gradient descent on full-dimensional factorization converges to the minimum nuclear norm solution."*

In a follow-up work, however, Razin & Cohen (2020) presented a counter-example demonstrating that implicit regularization in BM factorization cannot be explained by minimal nuclear norm, or in fact any norm. Specifically, they showed that there are instances where the gradient method applied to BM factorization yields a diverging sequence, and all norms thus grow toward infinity. Intriguingly, despite this divergence, they found that the rank of the estimate decreases toward its minimum.

Although this phenomenon might seem surprising initially, it is not uncommon for diverging sequences to follow a structured path. A prime example is the Power Method, the fundamental algorithm for finding the largest eigenvalue and eigenvector pair of a matrix. Starting from a random initial point $x_0$, the Power Method iteratively updates the estimate by multiplying it with the matrix. This process amplifies the component of the vector that aligns with the direction of the dominant eigenvector more than the other components, progressively leading $x_k$ to align with this eigenvector. In practical implementations, $x_k$ is scaled after each iteration to avoid numerical issues from divergence.

This perspective serves as the foundational motivation guiding our approach. Specifically, our key insight is that the implicit regularization in BM factorization (and neural networks) is driven by divergent dynamical behavior. This is markedly different from the standard (convergent) optimization dynamics helping with the data fitting. In this context, we hypothesize that these forces do not merely coexist but actively compete, influencing model behavior and performance in fundamentally conflicting ways. Our main goal in the development of this paper is to devise an approach that unravels these competing forces.

## 1.1 Overview of main contributions

- **A novel formulation for matrix factorization**. We model $X = UDV^\top$, where $U$ and $V$ are constrained within Frobenius norm balls. Projection onto this ball results in a scaling step similar to the Power Method. The middle term $D$ is a diagonal matrix that allows the model to explore the entire search space despite $U$ and $V$ being bounded.

  Empirically we demonstrate that the gradient method applied to the proposed formulation exhibits a pronounced implicit bias toward low-rank solutions. We compare our formulation against standard BM factorization with two unconstrained factors. Specifically, we investigate key factors such as step size and initialization, which prior work suggests might be contributing to implicit bias. We find that our factorization approach largely obviates the need to rely on these conditions: it consistently finds truly (rather than approximately) low-rank solutions across a wide range of initializations and step-sizes in our experiments. We believe these findings should be of broader interest to research on implicit bias.

- **A novel neural network architecture**. Motivated by the strong bias for low-rank solutions of the proposed factorization, we subsequently extend it to deep neural networks. Specifically, we do so by adding constrained layers and diagonal components. We demonstrate numerically that this constrained model performs on par with ReLU activations across various regression and classification tasks, without requiring further nonlinear activations. Importantly, even here, our approach exhibits bias towards low-rank solutions, resulting in a natural pruning procedure to obtain compact, lightweight networks without compromising performance.

## 2 Related Work

**Burer-Monteiro factorization.** BM factorization is initially proposed for solving semi-definite programs (Burer & Monteiro, 2003a; 2005; Boumal et al., 2016; 2020; Cifuentes, 2021) and is recognized for its efficiency in addressing low-rank matrix optimization problems, see (Sun & Luo, 2016; Bhojanapalli et al., 2016; Park et al., 2017; 2018; Hsieh et al., 2018; Sahin et al., 2019; Lee et al., 2022; Yalçın et al., 2023) and the references therein. Building on the connections between training problems for (non-linear) two-layer neural networks and convex (copositive) formulations (see (Pilanci & Ergen, 2020; Ergen & Pilanci, 2020; Sahiner et al., 2021) and the references therein), Sahiner et al. (2023) recently introduced BM factorization to solve convex formulations for various neural network architectures, including fully connected networks with ReLU and gated ReLU (Fiat et al., 2019) activations, two-layer convolutional neural networks (LeCun et al., 1989a), and self-attention mechanisms based on transformers (Vaswani et al., 2017).

**Implicit regularization.** While neural network-based systems are rapidly emerging as a dominant technology, the mechanisms underlying their generalization capabilities are yet to be fully understood. One promising line of research aims to explain this success through the concept of 'implicit regularization,' which is induced by the optimization methods and formulations used during neural network training (Neyshabur et al., 2014; 2017; Neyshabur, 2017). Several studies have explored matrix factorization to investigate implicit bias in linear neural networks (Gunasekar et al., 2017; Arora et al., 2018; Razin & Cohen, 2020; Belabbas, 2020; Li et al., 2021). Le & Jegelka (2022) extended these results to the final linear layers of nonlinear ReLU-activated feedforward networks with fully connected layers and skip connections. More recently Timor et al. (2023) investigated implicit regularization in ReLU networks. Much of the existing work focuses on gradient flow dynamics in the limit of infinitesimal learning rates. In particular, Gidel et al. (2019) examined discrete gradient dynamics in two-layer linear neural networks, showing that the dynamics progressively learn solutions of reduced-rank regression with a gradually increasing rank.

**Constrained neural networks.** Regularizers are frequently used in neural network training to prevent overfitting and improve generalization (Goodfellow et al., 2016), or to achieve structural benefits such as sparse and compact network architectures (Scardapane et al., 2017). However, it is conventional to apply these regularizers as penalty functions in the objective rather than as hard constraints, addressing them within gradient-based optimization via their (sub)gradients. This approach is likely favored due to the ease of implementation, as pre-built functions are readily available in common neural network packages. Several independent studies have also applied proximal methods across different networks and applications, finding that proximal-based methods tend to yield solutions with more pronounced structures (Bai et al., 2019; Yang et al., 2020; Yun et al., 2021; Yurtsever et al., 2021; Yang et al., 2022). Structural regularization in the form of hard constraints, however, appears to be rare in neural network training. One notable exception is in the context of neural network training with the Frank-Wolfe algorithm (Pokutta et al., 2020; Zimmer et al., 2022; Macdonald et al., 2022), where constraints are necessary due to the algorithm's requirement for a bounded domain.

There are other use cases of constraints in neural networks. For instance, constraints can be applied to ensure adherence to real-world conditions in applications where they are necessary (Pathak et al., 2015; Márquez-Neila et al., 2017; Jia et al., 2017; Kervadec et al., 2019; Weber et al., 2021), or to incorporate physical laws in physics-informed neural networks (Raissi et al., 2019; Lu et al., 2021; Patel et al., 2022). In these cases, the feasible set is typically complex and difficult to project onto; therefore, optimization algorithms like primal-dual methods or augmented Lagrangian techniques are used. Constraints are also present in lifted neural networks, a framework where the training problem is reformulated in a higher-dimensional 'lifted' space. In this space, conventional activation functions like ReLU, used in the original formulation, are expressed as constraints (Askari et al., 2018; Sahiner et al., 2021; Bartan & Pilanci, 2021).

**Pruning.** Neural networks are often trained in an over-parameterized regime to enhance generalization and avoid getting stuck in poor local minima. However, these models can suffer from excessive memory and computational resource demands, making them less efficient for deployment in real-world applications (Chang et al., 2021). Several compression techniques have been studied in the literature, including parameter quantization (Krishnamoorthi, 2018), knowledge distillation (Gou et al., 2021), and pruning.

Pruning reduces the number of parameters, resulting in more compact and efficient models that are easier to deploy. The literature on pruning is extensive, with diverse methods proposed with distinct characteristics. Various criteria are used to determine which weights to prune, including second-order derivative-based methods (LeCun et al., 1989b; Hassibi & Stork, 1992), magnitude-based pruning (Janowsky, 1989; Han et al., 2015), saliency heuristics (Mozer & Smolensky, 1988; Lee et al., 2018), and matrix or tensor factorization-based techniques (Xue et al., 2013; Sainath et al., 2013; Jaderberg et al., 2014; Lebedev et al., 2015; Swaminathan et al., 2020), among others. A comprehensive review of pruning methods is beyond the scope of this paper due to space limitations and the diversity of approaches. For more detailed reviews, we refer to (Reed, 1993; Blalock et al., 2020; Cheng et al., 2024), and the references therein.

## 3 MATRIX FACTORIZATION WITH A DIAGONAL COMPONENT

Consider the *matrix sensing* problem, where the goal is to recover a *positive semidefinite* (PSD) matrix $X \in \mathbb{S}_+^{d \times d}$ from a set of linear measurements $b = \mathcal{A}(X) \in \mathbb{R}^n$. We define $\mathcal{A} : \mathbb{R}^{d \times d} \to \mathbb{R}^n$ through symmetric measurement matrices $A_1, \ldots, A_n \in \mathbb{S}^{d \times d}$, such that $\mathcal{A}(X) = [\langle A_1, X \rangle \cdots \langle A_n, X \rangle]^\top$ and $\mathcal{A}^\top y = \sum_{i=1}^n y_i A_i$. We particularly focus on the data-scarce setting where $n \ll d^2$. A popular example is the matrix completion problem, where the goal is to reconstruct a matrix $X$ from a subset of its entries. This problem is inherently under-determined; however, successful recovery is possible if $X$ is known to be low-rank (Candes & Recht, 2012).

**Remark 1.** We focus on the recovery problem of a PSD matrix for simplicity; note that a non-square matrix sensing problem can also be reformulated as a PSD matrix sensing problem through a simple transformation (Park et al., 2017).

The problem described above can be cast as the following rank-constrained optimization problem:

$$\min_{X \in \mathbb{S}_+^{d \times d}} \quad f(X) := \frac{1}{2} \|\mathcal{A}(X) - b\|_2^2 \quad \text{subj. to} \quad \text{rank}(X) \leq r. \tag{1}$$

Although rank-constrained matrix optimization problems are typically NP-hard, various methods have been developed to provide practical approximations. Examples include hard thresholding algorithms (Jain et al., 2010; Goldfarb & Ma, 2011; Kyrillidis & Cevher, 2014), convex relaxation techniques (Candes & Recht, 2012; Recht et al., 2010), and BM factorization (Burer & Monteiro, 2003a; Sun & Luo, 2016; Bhojanapalli et al., 2016; Park et al., 2017).

The main idea behind BM factorization is to reparametrize the decision variable $X$ as $UU^\top$, where the factor $U \in \mathbb{R}^{d \times r}$, and $r$ is a positive integer that controls the rank of the resulting product. Problem (1) can then be reformulated as:

$$\min_{U \in \mathbb{R}^{d \times r}} \quad \frac{1}{2} \|\mathcal{A}(UU^\top) - b\|_2^2. \tag{2}$$

Despite the fact that finding the global minimum of (2) remains challenging, a local solution can be approximated using the gradient method (Lee et al., 2016). Initializing at $U_0 \in \mathbb{R}^{d \times r}$, perform the iteration:

$$U_{k+1} = U_k - \eta \nabla_U f(U_k U_k^\top), \tag{3}$$

where $\eta > 0$ is the step-size, and the gradient $\nabla_U f$ is computed as

$$\nabla_U f(UU^\top) = 2\nabla f(UU^\top)U = 2\mathcal{A}^\top(\mathcal{A}(UU^\top) - b)U.$$

Selecting the factorization rank $r$ is a critical decision that might impact the solution. A small $r$ may lead to spurious local minima, resulting in inaccurate outcomes (Waldspurger & Waters, 2020). Conversely, a large $r$ might weaken rank regularization, rendering the problem underdetermined. Conventional wisdom in BM factorization suggests finding a moderate choice that balances these two extremes. However, a key observation in (Gunasekar et al., 2017) is that the gradient method applied to (2) exhibits a tendency towards approximately low-rank solutions (*i.e.,* a decaying singular value spectrum) even when $r = d$. Below, we restate their conjecture:

**Conjecture in (Gunasekar et al., 2017).** Suppose gradient flow (*i.e.,* gradient descent with an infinitesimally small step-size) is initialized at a *full-rank matrix arbitrarily close to the origin*. If the limit of the gradient flow, $X_{\text{GF}} = UU^\top$, exists and is a global optimum of (1) with $\mathcal{A}(X_{\text{GF}}) = b$, then $X_{\text{GF}}$ is the minimal nuclear-norm solution to (1).

**Remark 2.** This conjecture outlines three conditions for implicit bias: a small step-size, near-origin full-rank initialization, and consistent measurements $(\mathcal{A}, b)$. However, these conditions are often unconventional. For example, small step-sizes are typically avoided because they slow down convergence. Similarly, near-origin initialization is counterintuitive, as the origin is a trivial spurious stationary point of (2).

### 3.1 THE PROPOSED FACTORIZATION

We propose reparameterizing $X = UDU^\top$, where $U \in \mathbb{R}^{d \times r}$ is constrained to have a bounded norm, and $D \in \mathbb{R}^{r \times r}$ is a non-negative diagonal matrix:

$$\min_{\substack{U \in \mathbb{R}^{d \times r} \\ D \in \mathbb{R}^{r \times r}}} \frac{1}{2} \|\mathcal{A}(UDU^\top) - b\|_2^2 \quad \text{s.t.} \quad \|U\|_F \leq \alpha, \ D_{ii} \geq 0, \ D_{ij} = 0, \ \forall i \text{ and } \forall j \neq i, \tag{4}$$

where $\alpha > 0$ is a model parameter. When the problem is well-scaled, for instance through basic preprocessing with data normalization, we found that $\alpha = 1$ is a reasonable choice.

Placing in multiple factors and with constraints, we perform projected-gradient updates on $U$ and $D$ with step-size $\eta > 0$:

$$\begin{aligned} U_{k+1} &= \Pi_U \left( U_k - \eta \nabla_U f(U_k D_k U_k^\top) \right) \\ D_{k+1} &= \Pi_D \left( D_k - \eta \nabla_D f(U_k D_k U_k^\top) \right), \end{aligned} \tag{5}$$

where $\Pi_U$ and $\Pi_D$ are projections for the constraints in (4); while the gradients are

$$\begin{aligned} \nabla_U f(UDU^\top) &= 2\nabla f(UDU^\top)UD, \\ \nabla_D f(UDU^\top) &= U^\top \nabla f(UDU^\top)U. \end{aligned}$$

## 3.2 NUMERICAL EXPERIMENTS ON MATRIX FACTORIZATION

We present numerical experiments comparing the empirical performance of the proposed approach with the classical BM factorization. Specifically, we examine the impact of initialization and step-size on the singular value spectrum of the resulting solution.

We set up a synthetic matrix completion problem aimed at recovering a PSD matrix $X_\natural = U_\natural U_\natural^\top$, where the entries of $U_\natural \in \mathbb{R}^{100 \times 3}$ are independently and identically (*iid*) drawn from the standard Gaussian distribution. We randomly sample $n = 900$ entries of $X_\natural$ and store them in the vector $b \in \mathbb{R}^n$. The goal is to recover $X_\natural$ from $b$ by solving problems (2) and (4). For initialization, we generate $U_0 \in \mathbb{R}^{d \times d}$ with entries drawn *iid* from standard Gaussian distribution and rescale it to have the Frobenius norm $\xi > 0$ (we investigate the impact of $\xi$). We initiate $D_0$ from the identity matrix.

The results are shown in Figure 1. First, we examine the impact of step-size. To this end, we fix $\xi = 10^{-2}$ and test different values of $\eta$. In the left panel, we plot the objective residual as a function of iterations. As expected, we observe that a smaller step-size slows down convergence. In the right panel, we plot the singular value spectrum of the results attained after $10^6$ iterations. We observe no direct connection between step-size and implicit bias in BM factorization.

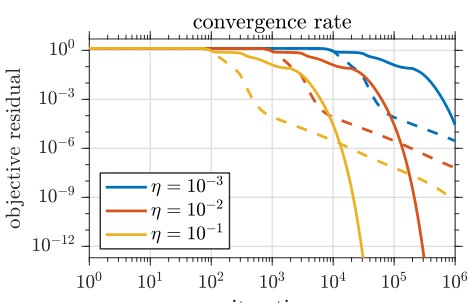
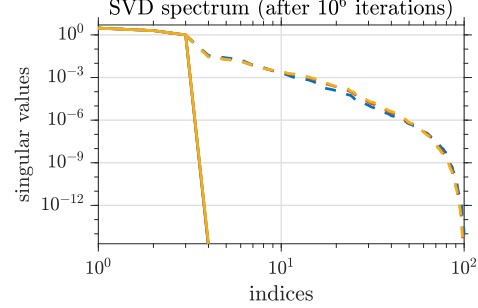

Impact of **step-size** ($\eta$), in **noiseless** setting, with fixed initialization.

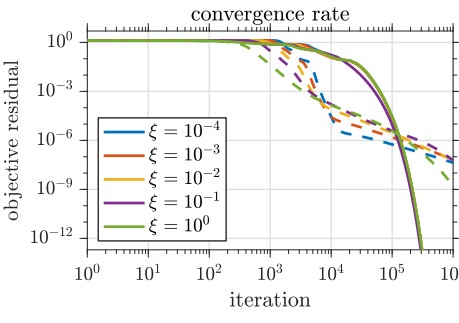
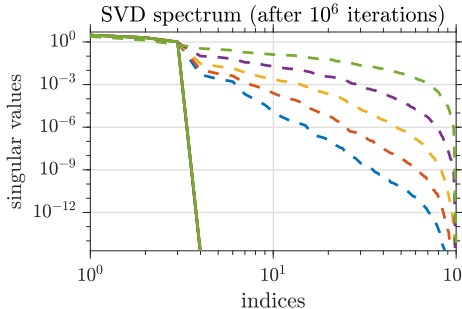

Impact of **initial distance to origin** ($\xi$), in **noiseless** setting, with fixed step-size.

Figure 1: Impact of step-size and initialization on implicit bias. **Solid lines represent our UDU factorization**, while **dashed lines denote the classical BM factorization**. [*Left*] Objective residual vs. iterations. [*Right*] Singular value spectrum after $10^6$ iterations. In all cases, UDU produces truly low-rank solutions, whereas the classical approach results in approximate low-rank structures.

Next, we investigate the impact of initialization. We fix the step-size at $\eta = 10^{-2}$ and evaluate the effect of varying $\xi$. We observe a correlation between the implicit bias of the BM factorization and $\xi$, which determines the initial distance from the origin. Initializing closer to the origin in the classical BM factorization yields solutions with a faster spectral decay. Notably, the UDU factorization demonstrates a strong implicit bias toward truly low-rank solutions, regardless of the choice of $\eta$ or $\xi$.

We provide additional experiments on matrix factorization problems in the supplementary material. Specifically, Appendix A.1 considers the matrix completion problem with noisy measurements, where $b$ is perturbed with Gaussian noise. The results remain consistent with the noiseless case: the UDU

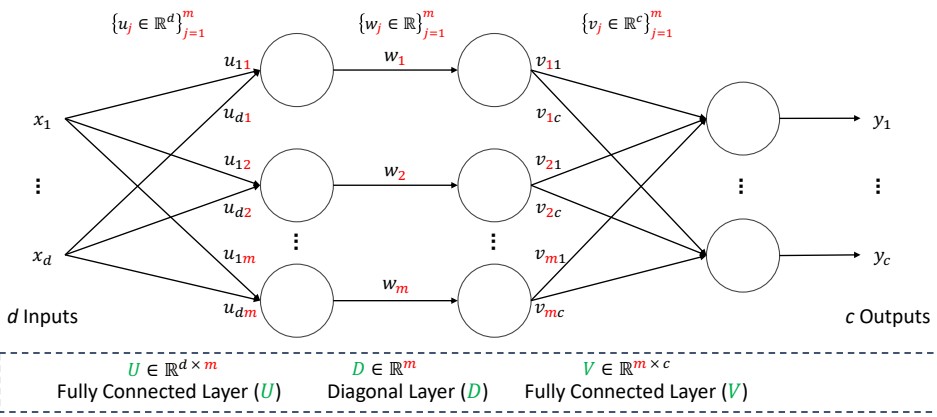

Figure 2: UDV structure. The weights in diagonal layer $D$ are denoted as $w_j$.

model exhibits an implicit bias toward truly low-rank solutions, while the classical BM factorization yields approximately low-rank solutions, with spectral decay strongly influenced by initialization.

In Appendix A.2, we present the evolution of the factors $U$ and $D$ over the iterations. Interestingly, the UDU model produces rank-revealing solutions, with the columns of $U$ converging to zero in certain directions. Specifically, $U$ tends to grow along certain directions, while the rescaling induced by projection onto the bounded constraint shrinks other coordinates. This behavior aligns with the mechanism of the power method and offers insights into its connection with divergent forces.

Additionally, we present numerical experiments on a matrix sensing problem arising in phase retrieval image recovery in Appendix A.3. As before, the UDU framework consistently promotes low-rank solutions, and this structural bias significantly enhances the quality of the recovered image, as demonstrated by our results.

## 4 FEEDFORWARD NEURAL NETWORKS WITH DIAGONAL HIDDEN LAYERS

This section extends our approach to neural networks. Consider a dataset comprising $n$ data points $(\mathbf{x}_i, \mathbf{y}_i) \in \mathbb{R}^d \times \mathbb{R}^c$. We first define a three-layer neural network defined as

$$\phi(\mathbf{x}) := \sum_{j=1}^{m} \mathbf{v}_j w_j \mathbf{u}_j^\top \mathbf{x} \approx \mathbf{y}. \tag{6}$$

The first and third layers are fully connected, and the middle is a diagonal layer, as illustrated in Figure 2. Drawing parallels between our matrix factorization model in (4) and neural network training, we impose Euclidean norm constraints on the weights of the fully connected layers. Under these conditions, the training problem can be formulated as follows:

$$\min_{\substack{\mathbf{u}_j \in \mathbb{R}^d \\ \mathbf{v}_j \in \mathbb{R}^c \\ w_j \in \mathbb{R}}} \quad \frac{1}{2n} \sum_{i=1}^{n} \| \sum_{j=1}^{m} \mathbf{v}_j \, w_j \, \mathbf{u}_j^\top \mathbf{x}_i - \mathbf{y}_i \|_2^2$$

$$\text{subj.to} \quad \sum_{j=1}^{m} \|\mathbf{u}_j\|_2^2 \leq 1, \quad \sum_{j=1}^{m} \|\mathbf{v}_j\|_2^2 \leq 1, \quad \text{and} \quad w_j \geq 0; \quad \text{for all} \quad j = 1, \ldots, m. \tag{7}$$

We refer to this neural network structure as UDV.

**Remark 3.** The norm constraints in our training problem can be interpreted as a stronger form of weight decay, one of the most commonly used regularization techniques in neural networks, which lends further justification to our formulation.

**Remark 4.** In Appendix B.1, we discuss three design variations of the UDV model with slight modifications to the constraints.

Table 1: Initial model setting (structure is #input - #hidden - #output neurons)

| Dataset | HPART (Anna Montoya, 2016) | NYCTTD (Risdal, 2017) | MNIST (LeCun et al., 2010) |
|---|---|---|---|
| Structure ($d$-$m$-$c$) | 79 - 26 - 1 | 12 - 10 - 1 | MaxViT-T(Tu et al., 2022): 512 - 341 - 10
EfficientNet-B0(Tan & Le, 2019): 1280 - 853 - 10
RegNetX-32GF(Radosavovic et al., 2020): 2520 - 1680 - 10 |
| Constraints or Activation | | UDV (7) and the variants in appendix, ReLU, UV | |
| Optimizer | | Adam (Kingma & Ba, 2014), Mini-Batch Gradient Descent (MBGD (LeCun et al., 2002)),
NAdam (Dozat, 2016), Mini-Batch Gradient Descent with Momentum (MBGDM (Sutskever et al., 2013)) | |
| Learning Rate | [1e-4, 1e-3, 1e-2, 1e-1, 1*, 2*, 3*]
(*Not applied to UV since it failed to converge) | | Adam/NAdam: [1e-6, 1e-5, 1e-4, 1e-3, 1e-2, 1e-1, 1]
MBGD/MBGDM: [1e-3, 1e-2, 1e-1, 1, 2, 3, 5] |
| Batch Size | 128 | | MaxViT-T (128); EfficientNet-B0 (384); RegNetX-32GF (128) |
| Loss Function | ½ Mean Squared Error (MSE) | | Cross Entropy |
| #Epochs | 200 | 50 | 70 |
| #Seeds | 1000 | 100 | 1 |

## 4.1 NUMERICAL EXPERIMENTS ON NEURAL NETWORKS

### 4.1.1 LOW-RANK BIAS IN NEURAL NETWORK TRAINING

We tested the proposed UDV framework on both regression and classification tasks, comparing it with fully connected two-layer neural networks (denoted as UV in the subsequent text) using both linear activation and ReLU activation functions. This comparison is fair in terms of computational cost, as the cost incurred by the diagonal layer -which can also be viewed as a parametrized linear activation function- is negligible.

**Datasets.** We used two datasets for the regression tasks: House Prices - Advanced Regression Techniques (HPART) (Anna Montoya, 2016) and New York City Taxi Trip Duration (NYCTTD) (Risdal, 2017). We allocated 80% of the data for the training and reserved the remaining 20% for the validation. We select the number of hidden neurons in the diagonal layer as $m = \texttt{round}\big(\sqrt{(c+2)d} + 2\sqrt{d/(c+2)}\big)$, following (Huang, 2003).

The classification tasks were evaluated on the normalized MNIST dataset (LeCun et al., 2010). We applied transfer learning by replacing the classifier layers of three advanced neural networks –MaxViT-T (Tu et al., 2022), EfficientNet-B0 (Tan & Le, 2019), and RegNetX-32GF (Radosavovic et al., 2020)– with UDV, while using pre-trained weights from ImageNet-1K (Deng et al., 2009). Specifically, all layers before the first fully connected layer of the classifier were retained while the subsequent layers were replaced. The number of hidden neurons in the diagonal layer was selected as $m = \texttt{floor}\big(\frac{2}{3}d\big)$.

**Implementation details.** All classification tasks were conducted on the NVIDIA A100 GPU with four cores of the AMD Epyc 7742 processor, while regression tasks were conducted on a single core of the Intel Xeon Gold 6132 processor. We used Python version 3.9.5 and PyTorch version 2.0.1.

Table 1 summarizes the experimental setup, including network architectures, problem formulations, optimization algorithms, and parameters such as learning rates, batch sizes, number of epochs, and random seeds. UV and UDV models are initialized identically, with the initial $D$ chosen as the identity matrix. The results are averaged over the random seeds for robustness. The validation loss, used as a generalization metric in regression, is averaged over the final 20 epochs for the HPART dataset and the final 5 epochs for the NYCTTD dataset. Similarly, validation accuracy for classification tasks is averaged over the last 5 epochs to ensure stability in the reported values.

**Results and Discussions.** Table 2 presents the validation loss (for regression) or validation accuracy (for classification) of the UDV model compared to the classical UV model with linear and ReLU activation functions. For each configuration (dataset and model architecture), the results are obtained by selecting the best algorithm and learning rate pair. Moreover, Figure 3 illustrates the singular value spectrum of the solutions corresponding to each entry in these tables. We focus on the singular values from the $U$ and $UD$ layers, as they generate the primary data representation, while omitting the $V$ layer, which serves as the feature selection layer and is a tall matrix by definition, given that $c \ll m$ in most cases. Collectively, these results show that the UDV framework achieves competitive prediction accuracy while exhibiting a strong implicit bias toward low-rank solutions, as indicated by the faster decay in the singular value spectrum.

Table 2: Model performance using different models. M, E, and R represent the transferred models MaxVit-T, EfficientNet-B0, and RegNetX-32GF, respectively.

| Tasks | Regression (Test Loss) | | Classification (Test Accuracy) | | |
|---|---|---|---|---|---|
| Dataset | HPART | NYCTTD | MNIST | | |
| UDV | $1.304 \times 10^{-3}$ Adam: $10^{-3}$ | $5.248 \times 10^{-6}$ NAdam: $10^{-4}$ | M: 99.67% MBGDM: $10^{-2}$ | E: 99.63% MBGDM: $10^{-1}$ | R: 99.74% MBGDM: $10^{-2}$ |
| UV | $1.333 \times 10^{-3}$ Adam: $10^{-3}$ | $5.251 \times 10^{-6}$ Adam: $10^{-3}$ | M: 99.69% MBGDM: $10^{-2}$ | E: 99.60% Adam: $10^{-3}$ | R: 99.66% MBGD: $10^{0}$ |
| UV-ReLU | $1.167 \times 10^{-3}$ Adam: $10^{-3}$ | $5.323 \times 10^{-6}$ NAdam: $10^{-3}$ | M: 99.68% NAdam: $10^{-4}$ | E: 99.68% MBGDM: $10^{-1}$ | R: 99.73% MBGD: $10^{0}$ |

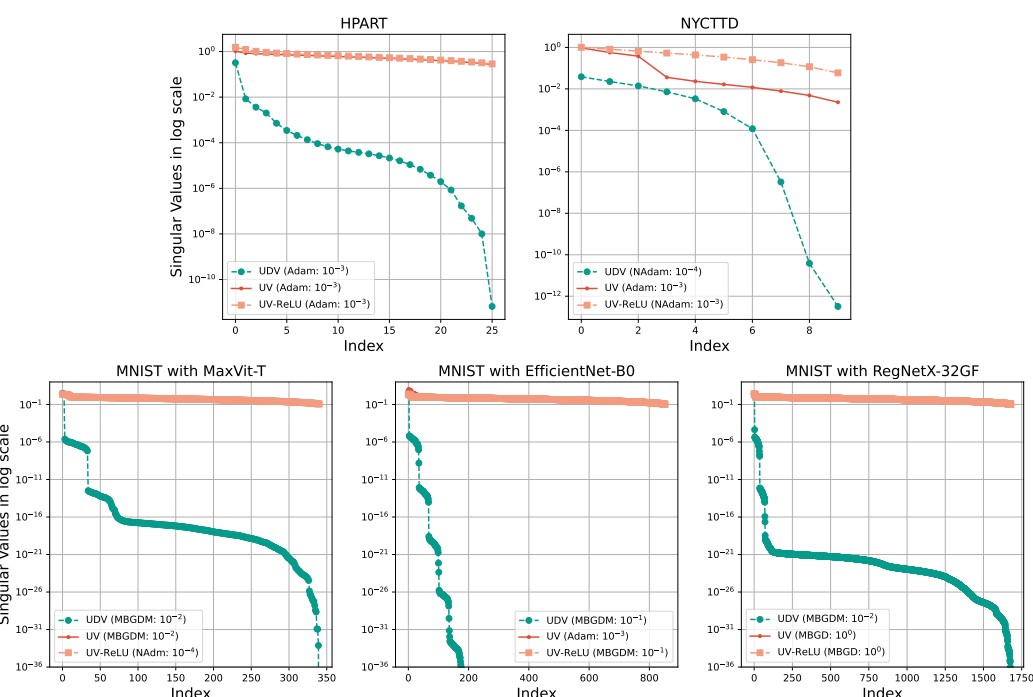

Figure 3: Singular value spectrum corresponding to the solutions reported in Table 2.

### 4.1.2 REDUCING NETWORK SIZE WITH SVD-BASED PRUNING

Efficient and lightweight feed-forward layers are crucial for real-world applications. For instance, the Apple Intelligence Foundation Models (Gunter et al., 2024) recently reported that pruning hidden dimensions in feed-forward layers yields the most significant gains in their foundation models. Building on this insight, we leverage the inherent low-rank bias of the UDV architecture through an SVD-based pruning strategy to produce compact networks without sacrificing performance.

A low-rank solution was observed when applying SVD to $UD$ layers:

$$UD = \text{USV}^\top, \quad \text{U} \in \mathbb{R}^{d \times m}, \quad \text{S} \in \mathbb{R}^{m \times m}, \quad \text{V}^\top \in \mathbb{R}^{m \times m} \tag{8}$$

By dropping small singular values in S, these matrices can be truncated to $\bar{\text{U}} \in \mathbb{R}^{d \times r}$, $\bar{\text{S}} \in \mathbb{R}^{r \times r}$ and $\bar{\text{V}}^\top \in \mathbb{R}^{r \times m}$, where $0 < r < m$. Consequently, $(m - r)$ neurons can be pruned, and new weight matrices are assigned:

$$\bar{U} = \bar{\text{U}} \in \mathbb{R}^{d \times r}, \quad \bar{D} = \bar{\text{S}} \in \mathbb{R}^{r \times r}, \quad \bar{V} = \bar{\text{v}}^T V \in \mathbb{R}^{r \times c}. \tag{9}$$

We applied this pruning strategy on the models from Table 2. The left part of Figure 4 presents an example comparing the generalization capability of pruned models. For comparison, we also

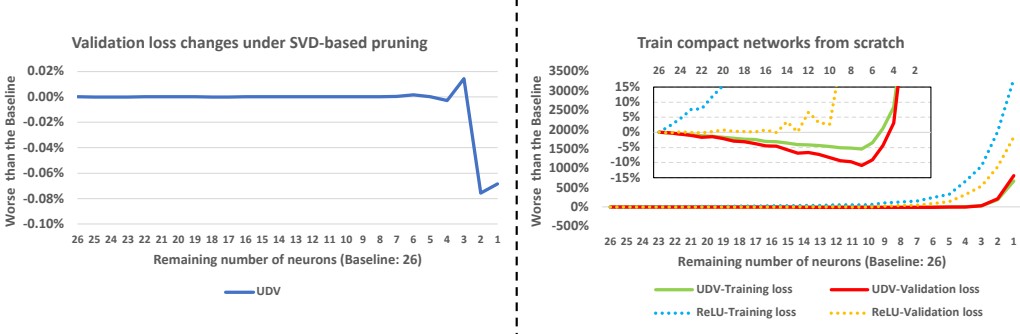

Figure 4: Comparison between SVD-based pruning vs. re-training compact networks (HPART dataset, NAdam algorithm with learning rate $10^{-3}$). Negative percentages indicate results that are better than the baseline. SVD-based pruning demonstrates that the UDV leads to a compact model without performance degradation, while retraining shows that the UDV achieves better generalization in a compact model.

created compact models by training from scratch with a reduced number of neurons $m$ in the hidden layer. The performance change for these models is shown in the right panel of Figure 4. Although our pruned networks derived from the UDV solution demonstrate that models with significantly fewer parameters can still achieve strong generalization, these compressed architectures are often more challenging to optimize directly within the reduced space, consistent with prior findings in the literature (Arora et al., 2018; Chang et al., 2021). We omit the results for retraining with the UV model, as they show similar trends to UDV in this context, though UDV generally exhibits superior generalization.

### 4.1.3    FURTHER DETAILS AND DISCUSSIONS

Our findings in experiments on neural networks align with the results observed in matrix factorization. A key distinction, however, was the use of different optimization algorithms, including stochastic gradients and momentum steps, in neural network experiments. Despite these differences, the UDV architecture consistently demonstrated a strong bias toward low-rank solutions. We provide further experiments and additional details in the supplementary material and summarize the key results here:

- When designing our network architecture, we considered four variants of UDV, each differing slightly in their constraints. We selected the version presented in equation (7), as it generally produces the most pronounced decay in the singular value spectrum. For completeness, the details of the other three variants are provided in Appendix B.1.

- Appendix B.2 provides additional results for the experiment described in Section 4.1.1. Specifically, we present results analogous to those in Table 2 and Figure 3 but focusing exclusively on the MBGDM algorithm. These results show similar trends, highlighting consistency across different methods. Additionally, comprehensive performance comparisons across all algorithms and models in Table 1, including the design variants in Appendix B.1, are available in Tables SM2 to SM5 in the supplementary material.

- Additional details and results on SVD-based pruning are presented in Appendix B.3. We show that the UDV framework consistently achieves low-rank solutions across various problem configurations. Furthermore, we analyze the effect of learning rate on the singular value spectrum, similar to the analysis in Figure 1, but applied to neural network experiments, confirming that the UDV framework produces low-rank solutions across a broad range of learning rates.

- Appendix B.4 extends the UDV framework by incorporating non-linear ReLU activation. Preliminary experiments with the UDV-ReLU model reveal low-rank solutions similar to those identified in the original UDV framework.

- Appendix B.5 presents an experiment comparing the UDV framework against two/three-layer fully connected blocks and a UDV block without the constraints. Given that prior work on the implicit bias in neural networks suggests that increasing depth enhances the bias towards low-rank

solutions (Arora et al., 2019; Feng et al., 2022), it is natural to ask whether the pronounced bias in the UDV framework is merely a result of adding a diagonal layer. The results indicate that the pronounced bias in the UDV framework cannot be attributed solely to depth, highlighting the critical role of explicit constraints.

- Appendix B.6 compares the spectral decay obtained by the UDV network with those from classical weight decay regularization in two- and three-layer networks. The results highlight that while weight decay regularization enhances singular value decay, it cannot replicate the results observed with the UDV model.

## 5 CONCLUSIONS

We proposed a new matrix factorization framework, inspired by the observation that implicit bias is driven by dynamics—potentially divergent—that are distinct from those leading to convergence of the objective function. This framework constrains the factors within Euclidean norm balls and introduces a middle diagonal factor to ensure the search space is not restricted. Numerical experiments demonstrate that this approach significantly strengthens the low-rank bias in the solution.

To explore the broader applicability of our findings, we designed an analogous neural network architecture with three layers, constraining the fully connected layers and adding a diagonal hidden layer, referred to as UDV. Extensive experiments show that the proposed UDV architecture achieves competitive performance compared to standard fully connected networks, while inducing a structured solution with a strong bias toward low-rank representations. Additionally, we explored the utility of this low-rank structure by applying an SVD-based pruning strategy, illustrating how it can be leveraged to construct compact networks that are more efficient for downstream tasks.

## ETHICS STATEMENT

The research presented in this paper complies with the ICLR Code of Ethics. Importantly, this study uses only randomly generated synthetic data and publicly available datasets, which, to the best of our knowledge, do not involve crowdsourcing, human subjects, or sensitive data.

## REPRODUCIBILITY STATEMENT

All datasets are cited and publicly available. The experimental settings and computational environment are described in both the main text and the appendix. The original code, along with detailed instructions, is provided in the supplementary material. Additionally, all random seeds are specified to ensure exact reproducibility.

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

## A  ADDITIONAL DETAILS ON MATRIX FACTORIZATION EXPERIMENTS

### A.1  MATRIX COMPLETION UNDER GAUSSIAN NOISE

We conducted similar experiments also using noisy measurements: Let $b^\natural = \mathcal{A}(X^\natural)$ represent the true measurements, and assume $b = b^\natural + \omega$, where $\omega \in \mathbb{R}^n$ is zero-mean Gaussian noise with a standard deviation of $\sigma = 10^{-2}\|b\|_2$. The results remain consistent with the noiseless case. $UDU^\top$ exhibits implicit bias toward truly low-rank solutions, while the classical BM factorization yields approximately low-rank solutions, with the spectral decay influenced by initialization.

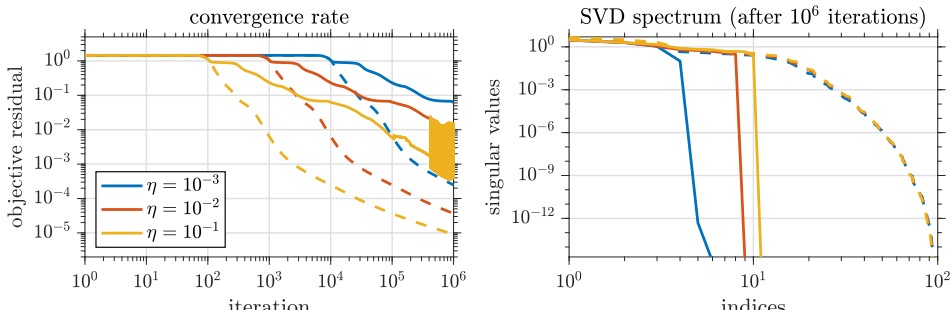

Impact of **step-size** ($\eta$), under **Gaussian noise**, with fixed initialization.

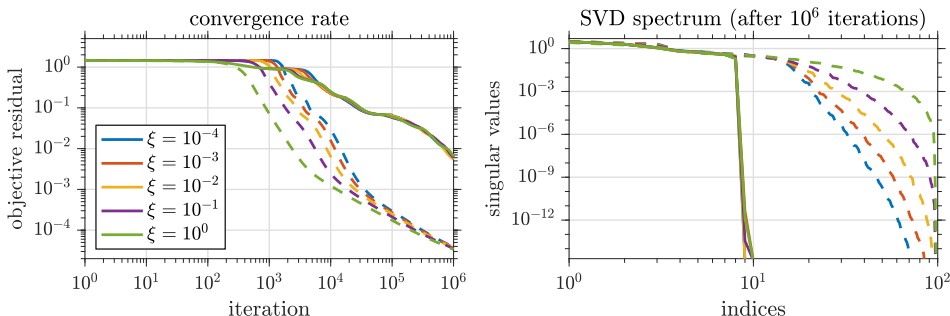

Impact of **initial distance to origin** ($\xi$), under **Gaussian noise**, with fixed step-size.

Figure SM1: Impact of step-size and initialization on implicit bias. **Solid lines represent our UDU factorization**, while **dashed lines denote the classical BM factorization**. [*Left*] Objective residual vs. iterations. [*Right*] Singular value spectrum after $10^6$ iterations.

### A.2  UDU FACTORIZATION PRODUCES RANK-REVEALING SOLUTIONS

The proposed factorization method naturally produces rank-revealing solutions. The columns of $U$ converge to zero in certain directions, resulting in truly low-rank solutions. Specifically, $U$ tends to grow along specific directions while the rescaling induced by projection onto the bounded constraint shrinks other coordinates. This behavior resembles the mechanism of the power method and provides insights into its connection with divergent forces.

In Figure SM4, we illustrate the evolution of the column norms of $U$ from the matrix completion experiment described in Section 3.2. Figure SM3 displays the corresponding entries in the diagonal factor $D$. For comparison, Figure SM4 shows the column norms of $U$ obtained from the same experiment using the standard BM factorization. The results are shown for the setting $\xi = 10^{-1}$ and $\eta = 10^{-1}$ over $10^5$ iterations, but the findings are qualitatively similar across other parameter settings.

We also repeat the experiment with the noisy measurements described in Appendix A.1. The results are shown in Figures SM5 to SM7.

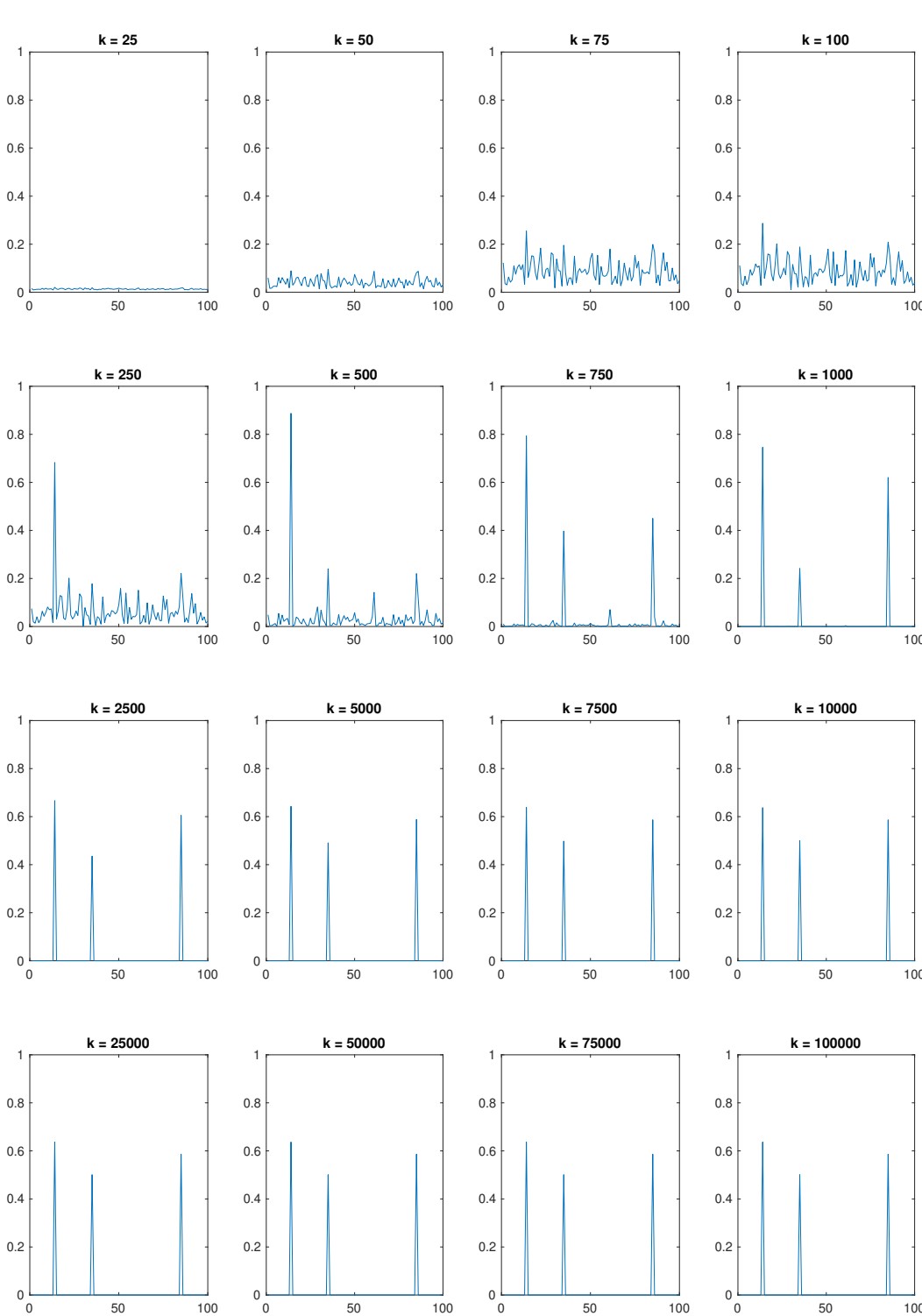

Figure SM2: Evolution of the column norms of $U$ during the matrix completion experiment using the UDU factorization. The x-axis represents column indices, and the y-axis shows the Euclidean norms of the columns.

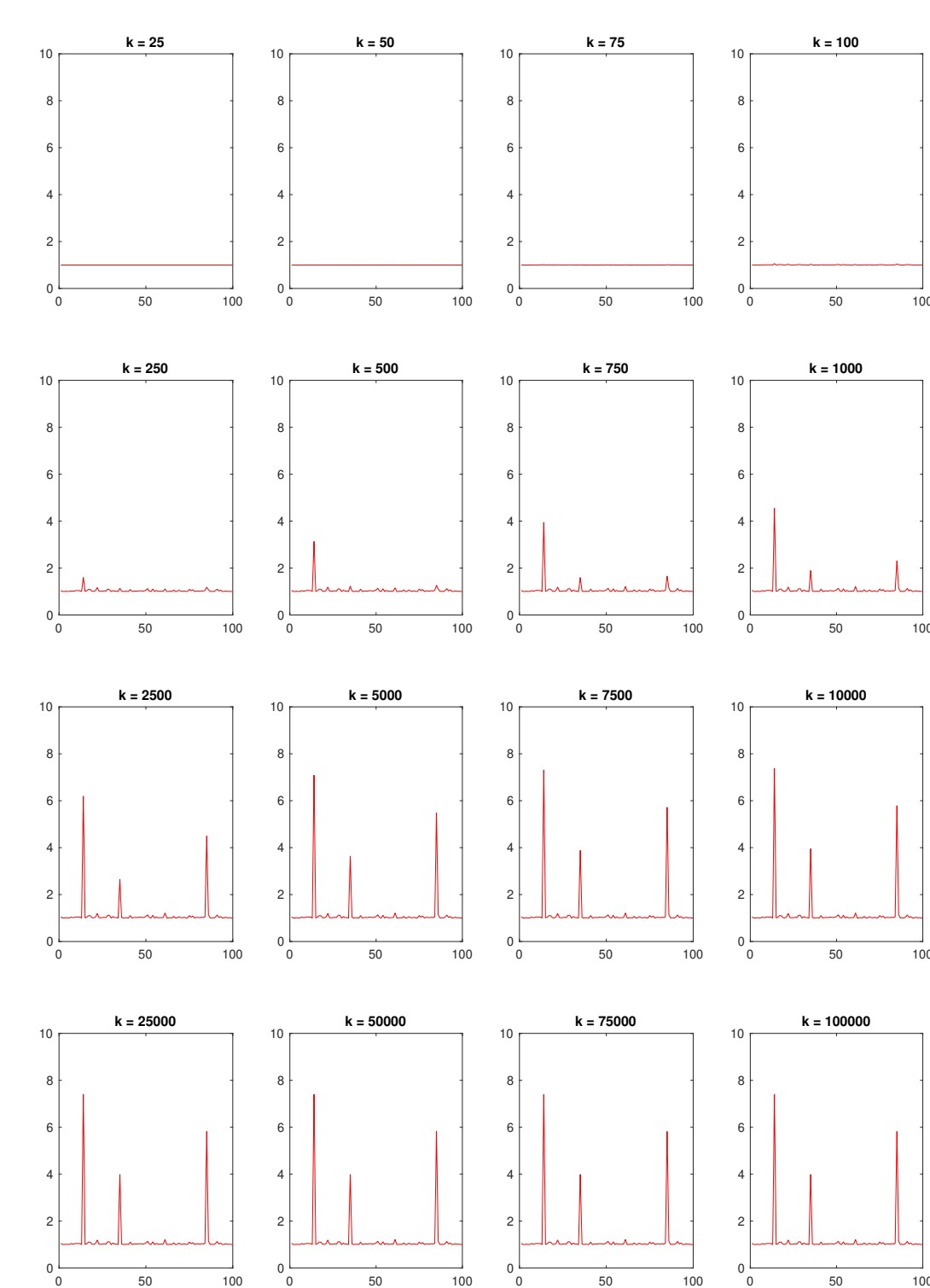

Figure SM3: Evolution of the diagonal entries of $D$ during the matrix completion experiment using the UDU factorization. The x-axis represents indices.

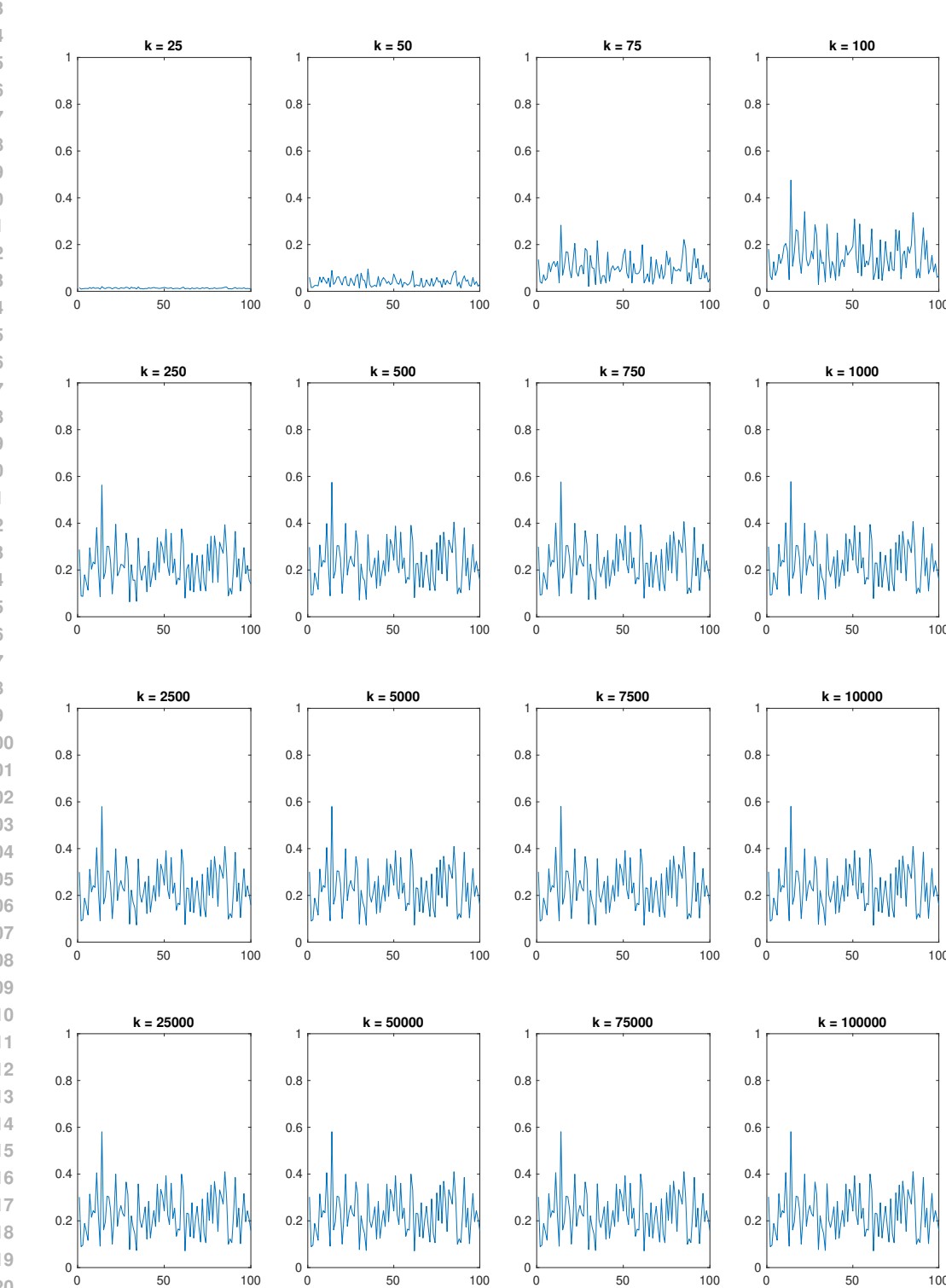

Figure SM4: Evolution of the column norms of $U$ during the matrix completion experiment using the standard BM factorization, shown for comparison. The x-axis represents column indices, and the y-axis shows the Euclidean norms of the columns.

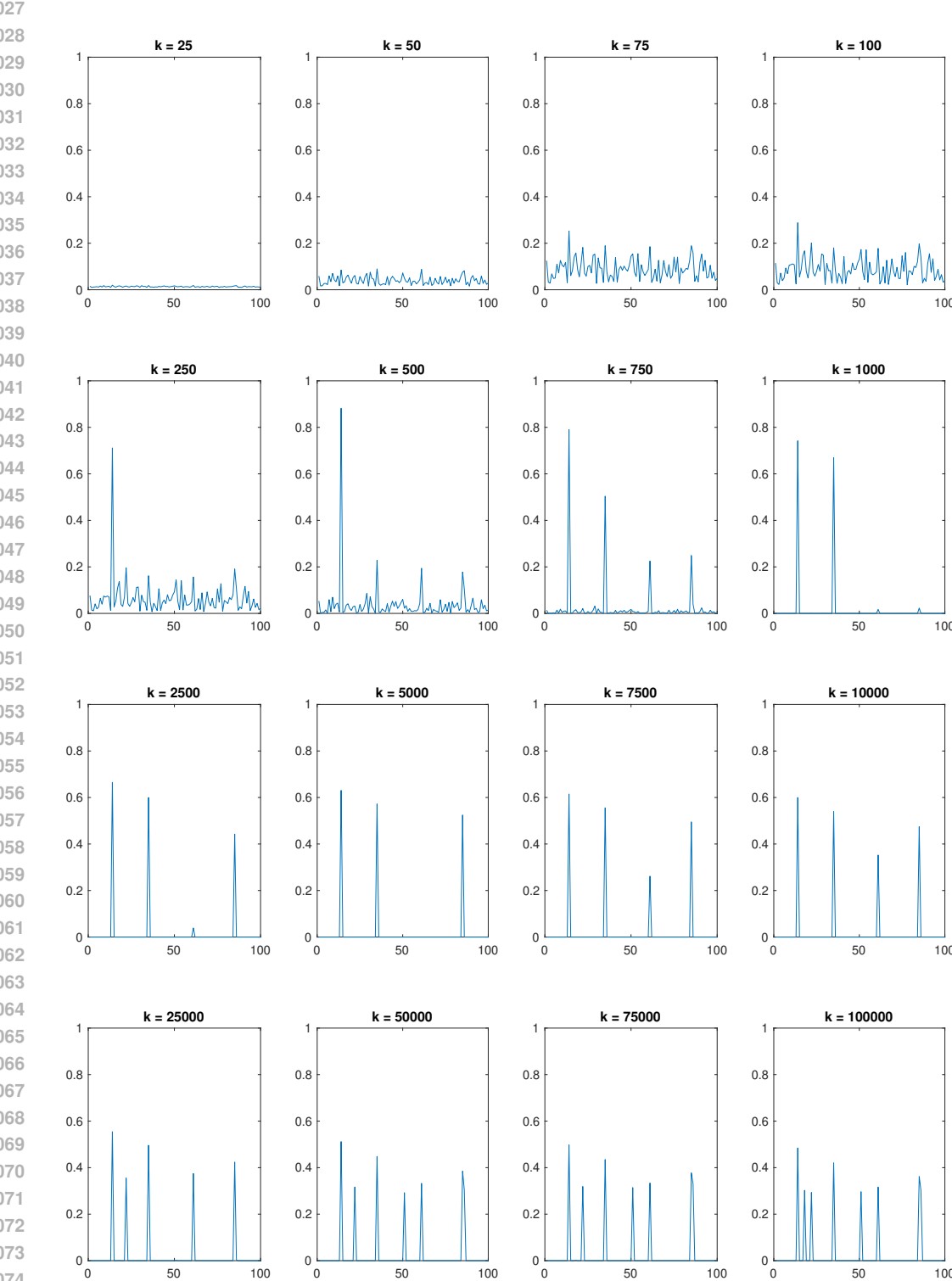

Figure SM5: Evolution of the column norms of $U$ during the matrix completion experiment using the UDU factorization with **noisy** measurements. The x-axis represents column indices, and the y-axis shows the Euclidean norms of the columns.

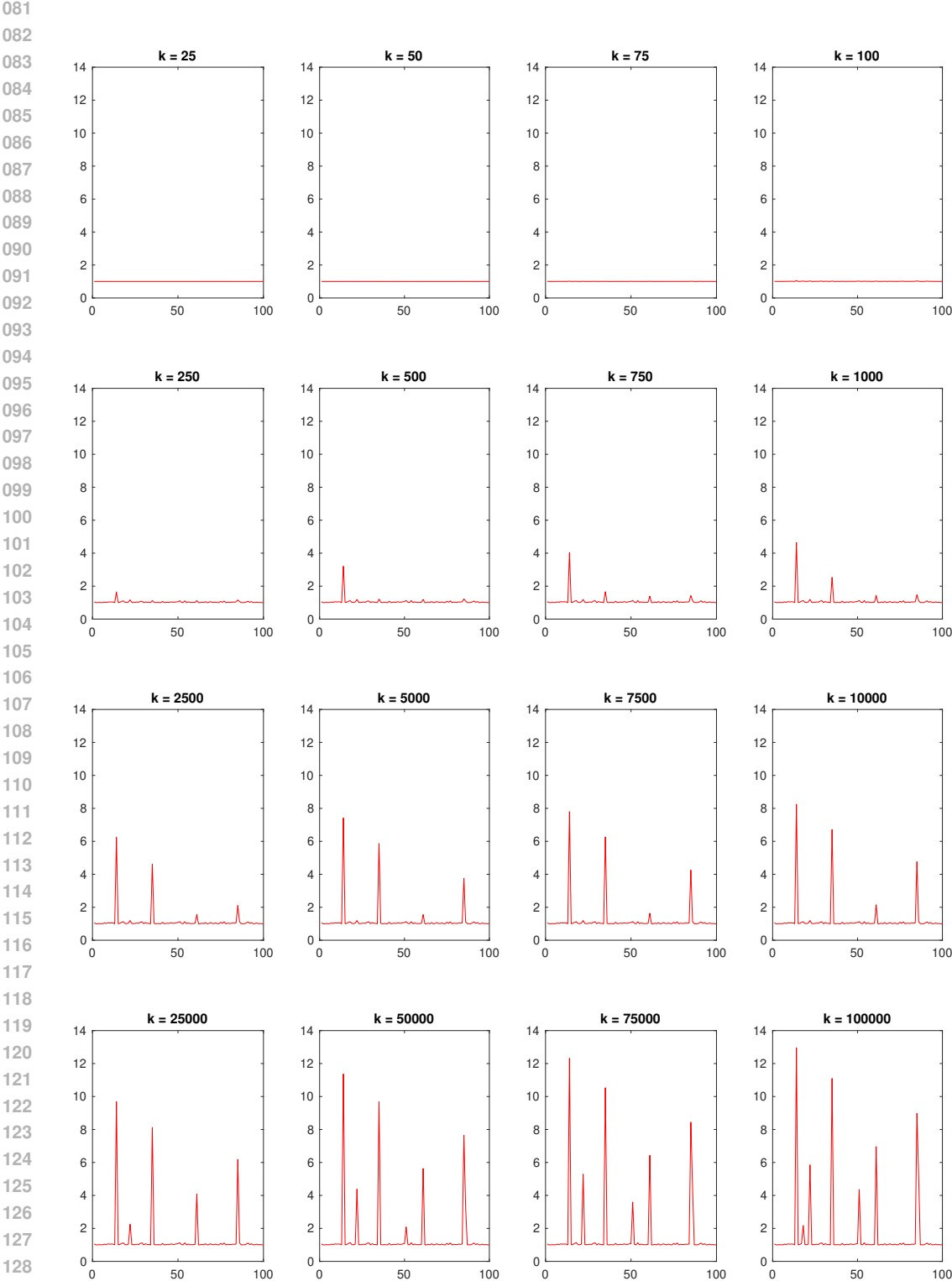

Figure SM6: Evolution of the diagonal entries of $D$ during the matrix completion experiment using the UDU factorization with **noisy** measurements. The x-axis represents indices.

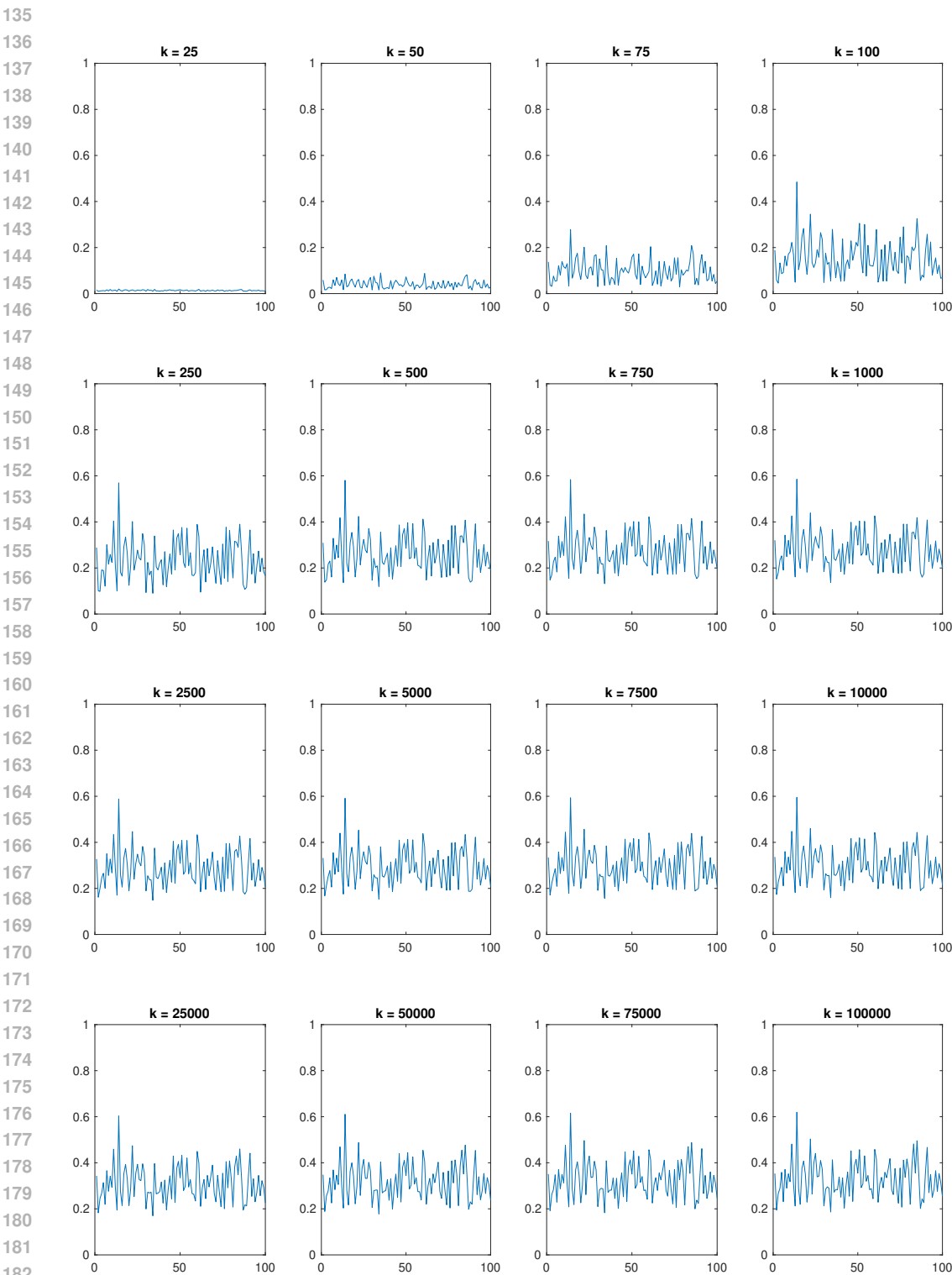

Figure SM7: Evolution of the column norms of $U$ during the matrix completion experiment using the standard BM factorization with **noisy** measurements, shown for comparison. The x-axis represents column indices, and the y-axis shows the Euclidean norms of the columns.

### A.3 PHASE RETRIEVAL IMAGE RECOVERY

We conducted a numerical experiment using the proposed matrix factorization model in (4) on the matrix sensing problem arising in phase retrieval image recovery (Candes et al., 2013). The objective is to recover a signal from its quadratic measurements of the form $y_i = |\langle a_i, x \rangle|^2$. Although the standard maximum likelihood estimators lead to a non-convex optimization problem due to the quadratic terms, the problem can be reformulated as minimizing a convex function under a rank constraint in a lifted space. By denoting the lifted variable as $X = xx^\top$, the measurements can be expressed as

$$y_i = \langle a_i^\top x, a_i^\top x \rangle = \langle a_i a_i^\top, xx^\top \rangle := \langle A_i, X \rangle,$$

which allows us to reformulate the problem as

$$\min_{X \in \mathbb{S}_+^{d \times d}} \quad \frac{1}{2} \|\mathcal{A}(X) - b\|_2^2 \quad \text{subj. to} \quad \text{rank}(X) \leq 1,$$

representing a special case of problem (1).

Specifically, we use a pre-processed gray-scale image of size $16 \times 16$ pixels, selected from the Pixel Art dataset (Elgazar, 2024), as the signal $x \in \mathbb{R}^n$ to recover, where $n = 256$. The image is vectorized and normalized. We generate a synthetic measurement system by sampling $a_1, \ldots, a_m$ from a standard Gaussian distribution, with $m = 2n$.

We then solved the problems (2) and (4). We used the following initializations: $U_0 \in R^{n \times n}$ was initialized with entries drawn *iid* from a standard Gaussian distribution and then rescaled to have a unit Frobenius norm. For $D_0 \in R^{n \times n}$, we used the identity matrix. The step size was set to $\eta = \frac{1}{L}$ where $L$ denotes the Lipschitz constant. After solving the problem, we recovered $x \in \mathbb{R}^n$ from the lifted variable $X \in \mathbb{R}^{n \times n}$ by selecting its dominant eigenvector.

Figure SM8 demonstrates that the proposed method consistently identifies low-rank solutions, in line with the results of our other experiments. Figure SM9 displays the recovered image, demonstrating that the proposed method achieves higher quality than BM factorization within the same number of iterations, which can be attributed to the low-rank structure of the solution.

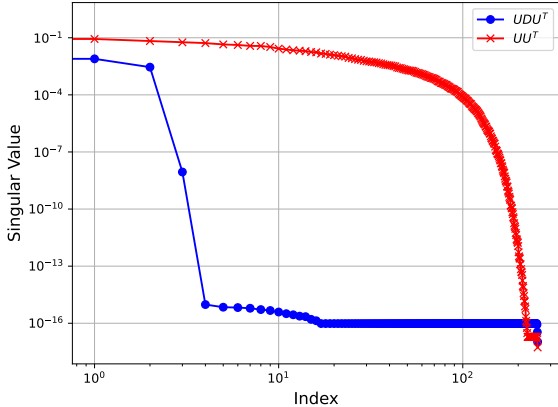

Figure SM8: Comparison of singular value spectrum in recovered image between methods based on $UDU^T$ and $UU^T$ reparameterization.

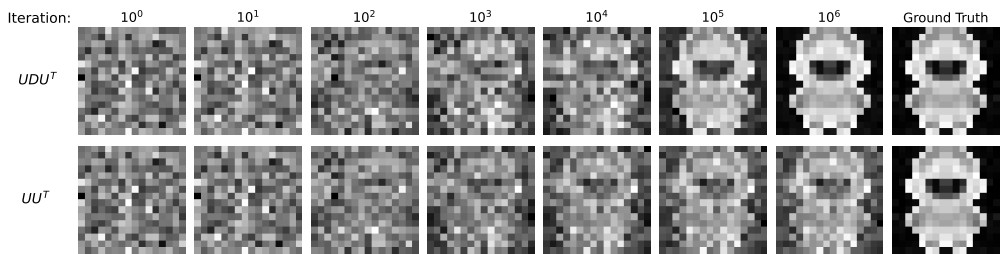

Figure SM9: Comparison of recovered image between methods based on $UDU^T$ and $UU^T$ reparameterization.

## B  ADDITIONAL DETAILS ON NEURAL NETWORK EXPERIMENTS

### B.1  DESIGN VARIANTS OF THE UDV ARCHITECTURE

When designing our network architecture, we considered four variants, including the one presented in equation (7) in Section 4. The three additional models were defined by the following sets of constraints:

$$\sum_{j=1}^{m} \|\mathbf{u}_j\|_2^2 \leq 1, \quad \sum_{j=1}^{m} \|\mathbf{v}_j\|_2^2 \leq 1 \qquad \text{(UDV-s)}$$

$$\|\mathbf{u}_j\|_2^2 \leq 1, \quad \|\mathbf{v}_j\|_2^2 \leq 1 \quad w_j \geq 0; \quad \text{for all } j = 1, \ldots, m \qquad \text{(UDV-v1)}$$

$$\|\mathbf{u}_j\|_2^2 \leq 1, \quad \|\mathbf{v}_j\|_2^2 \leq 1; \quad \text{for all } j = 1, \ldots, m \qquad \text{(UDV-v2)}$$

In detail, UDV-s is identical to UDV but omits the non-negativity constraints on the diagonal layer. UDV-v1, on the other hand, enforces row/column-wise norm constraints instead of the Frobenius norm used in UDV, while retaining the non-negativity constraints on the diagonal elements. Finally, UDV-v2 is identical to UDV-v1 but without the non-negativity constraints on the diagonal layer.

We only report results from UDV, as defined in equation (7), in the main text. However, we provide results for the other variants in the supplementary material for completeness and to illustrate the impact of different constraint settings on model performance.

Figure SM11 and Figure SM12 show that UDV consistently finds a low-rank solution. Generally, UDV exhibits the most pronounced decaying pattern in singular values. Additionally, we extended our experiments to include full-batch training on the HPART and NYCTTD datasets. Although full-batch training converges more slowly than stochastic (mini-batch) methods, it exhibits a similar singular value decay pattern, confirming our earlier observations.

### B.2  COMPARISON OF MODEL PERFORMANCE WITH THE MBGDM ALGORITHM

In Section 4.1, we presented the performance of each model (UDV, UV, and UV-ReLU) in terms of generalization power (measured by validation loss or validation accuracy) along with the corresponding singular value spectrum, as shown in Table 2 and Figure 3. The results were obtained by selecting the optimal algorithm and learning rate pair for each model. Here, we conduct the same experiment, but focus exclusively on the MBGDM algorithm. The results, presented in Table SM1 and Figure SM10 are similar to the ones in Section 4.1, show similar trends, highlighting consistency across methods.

### B.3  ADDITIONAL DETAILS ON THE PRUNING EXPERIMENT

The SVD-based pruning experiments yield conclusions consistent with the singular value decay pattern. UDV typically exhibits the fastest decay, leading to more compact models while maintaining performance, as shown in Figure SM13.

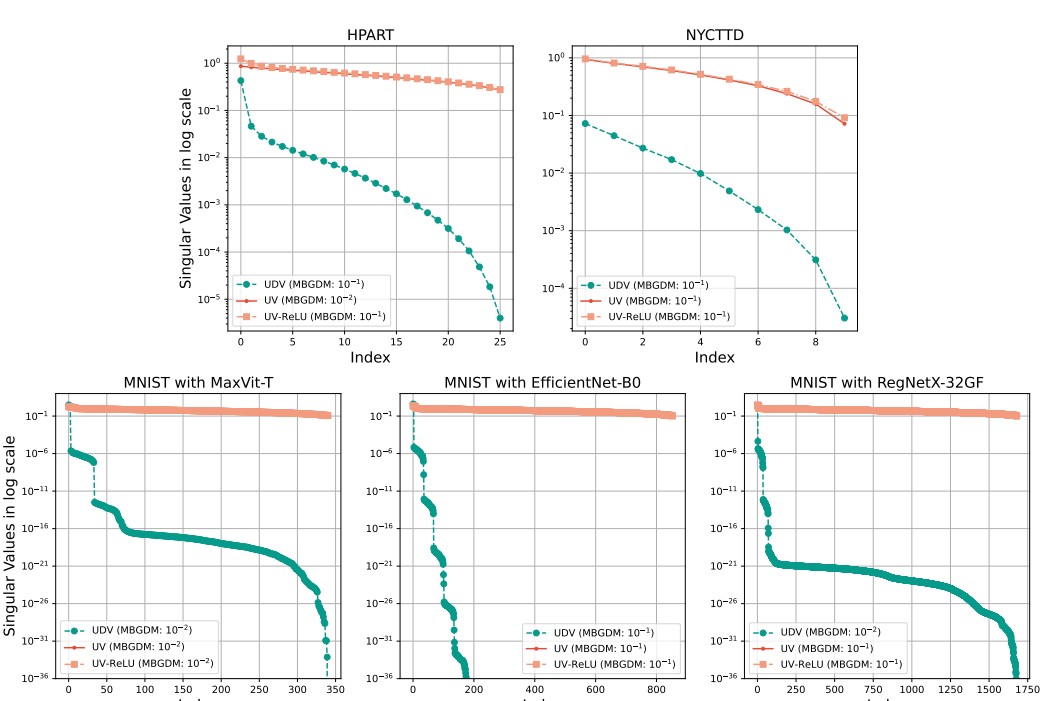

Figure SM10: Singular value spectrum corresponding to Table SM1.

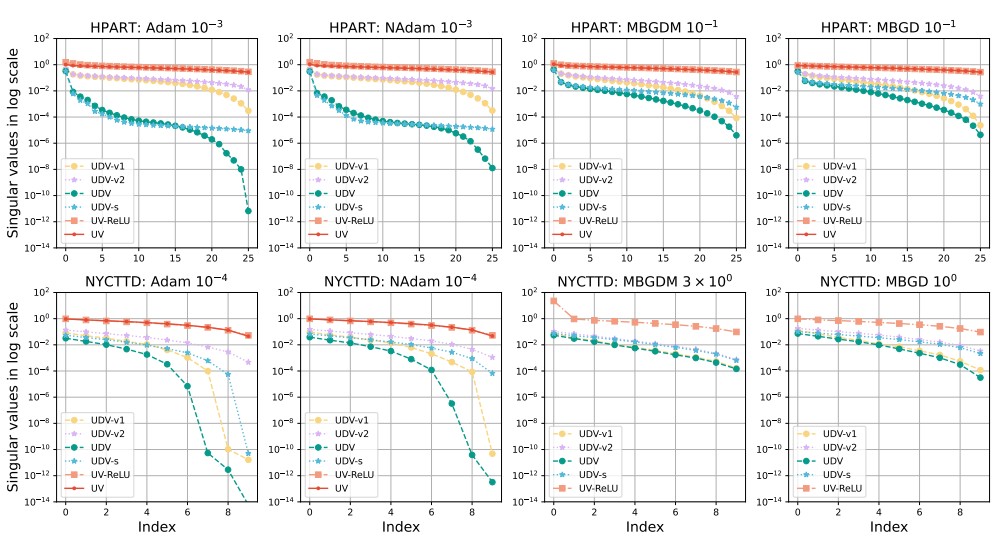

Figure SM11: Comparison of singular value pattern among all UDV variants, UV-ReLU and UV on the regression dataset.

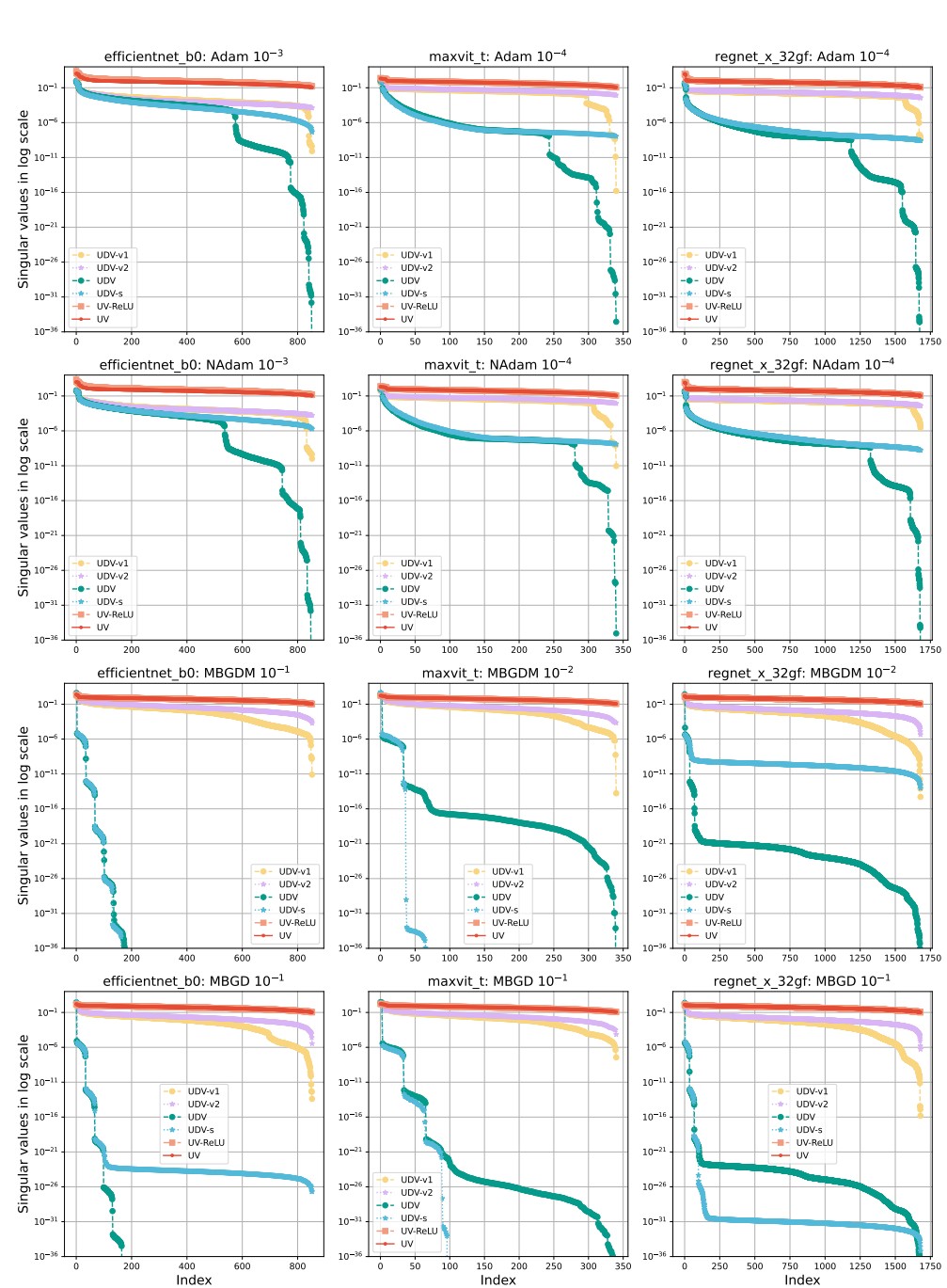

Figure SM12: Comparison of singular value pattern among all UDV variants, UV-ReLU and UV on the MNIST dataset.

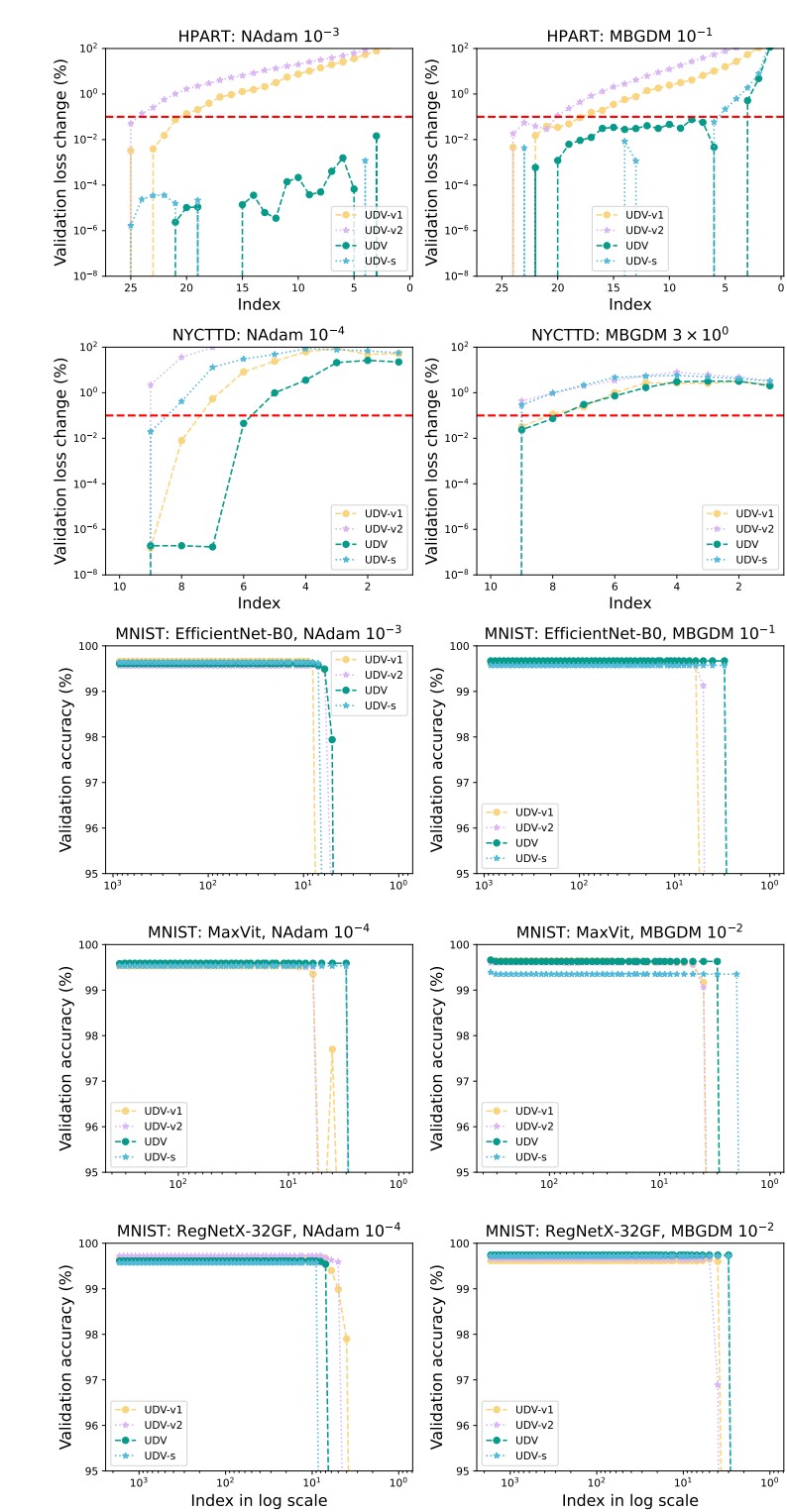

Figure SM13: The performance of SVD-based pruning. The index in x-axis represents the number of neurons in the diagonal layer after pruning. For the HPART and NYCTTD datasets, the validation loss change indicates how much worse the pruned model performs compared to the baseline (the model before pruning), expressed as a percentage ($\frac{loss_{pruned} - loss_{baseline}}{loss_{baseline}} \times 100\%$). Note that the pruned model may outperform the baseline, but negative values cannot be displayed on a logarithmic scale. The red dashed line denotes the 0.1% threshold, indicating negligible performance sacrifice. For the MNIST dataset, the results show the validation accuracy after pruning the model.

Table SM1: Model performance using MBGDM. M, E, and R represent the transferred models MaxVit-T, EfficientNet-B0, and RegNetX-32GF, respectively.

| Tasks | Regression (Test Loss) | | Classification (Test Accuracy) | | |
|---|---|---|---|---|---|
| Dataset | HPART | NYCTTD | MNIST | | |
| UDV | $1.345 \times 10^{-3}$ MBGDM: $10^{-1}$ | $5.248 \times 10^{-6}$ MBGDM: $10^{-1}$ | M: 99.67% MBGDM: $10^{-2}$ | E: 99.63% MBGDM: $10^{-1}$ | R: 99.74% MBGDM: $10^{-2}$ |
| UV | $1.337 \times 10^{-3}$ MBGDM: $10^{-2}$ | $5.259 \times 10^{-6}$ MBGDM: $10^{-1}$ | M: 99.69% MBGDM: $10^{-2}$ | E: 99.59% MBGDM: $10^{-1}$ | R: 99.56% MBGDM: $10^{-1}$ |
| UV-ReLU | $1.244 \times 10^{-3}$ MBGDM: $10^{-1}$ | $5.264 \times 10^{-6}$ MBGDM: $10^{-1}$ | M: 99.63% MBGDM: $10^{-2}$ | E: 99.68% MBGDM: $10^{-1}$ | R: 99.66% MBGDM: $10^{-1}$ |

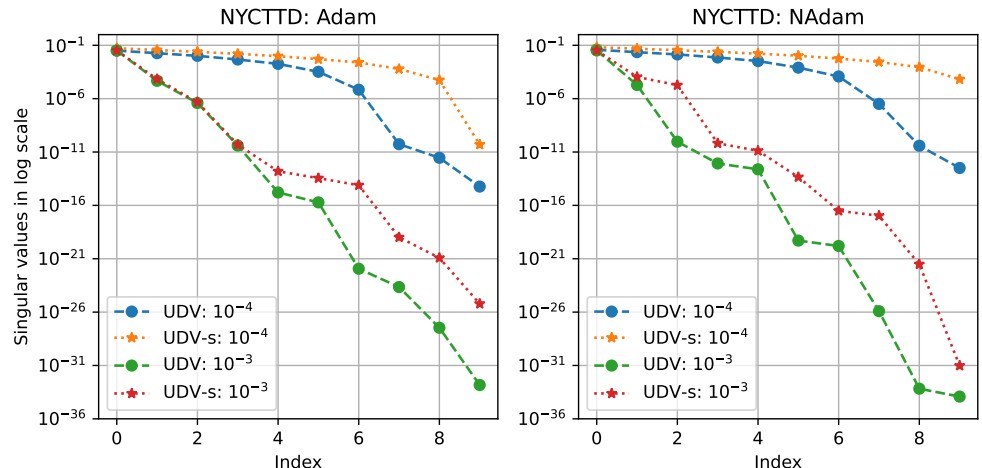

Figure SM14: Differences in singular value patterns across varying learning rates.

We observed that the learning rate can have some impact on the singular value decay pattern. In particular, a very large learning rate may cause oscillations in both training and validation loss, yet may result in a rapid decay of the spectrum. Conversely, a small learning rate may lead to a less pronounced spectral decay but still can yield a comparable validation loss to that of the optimal learning rates. For example, the difference in validation losses between the Adam optimizer with learning rates of 1e-3 and 1e-4 was negligible, yet the singular value spectra differed (see Figure SM14). This discrepancy also affects the performance of the pruning experiments (see Figure SM15).

When the learning rate is close to the optimal value, a "stair-step" pattern (see Figure SM16) can be observed on the loss or accuracy curve in the classification task. It could be attributed to the model continuously searching for a low-rank solution.

## B.4 INCORPORATING ReLU INTO UDV

To explore whether non-linear activation functions, particularly ReLU, can be used in our UDV framework, we conducted an additional experiment.

Building on the standard ReLU, we first introduced norm constraints, $\sum_{j=1}^{m} \|\mathbf{u}_j\|_2^2 \leq 1$ and $\sum_{j=1}^{m} \|\mathbf{v}_j\|_2^2 \leq 1$, to the layers both before and after ReLU, denoted as *ReLU(constrained)*. Next, we integrated ReLU into the diagonal layer of UDV. When the bounded $w_j$ is applied to the diagonal layer, we refer to the model as UDV-ReLU; otherwise, it is called UDV-ReLU-s.

Since the aim of this part is to verify the feasibility of incorporating ReLU into the UDV, we conducted a preliminary experiment, leaving a comprehensive study to future work. We derived the optimizers

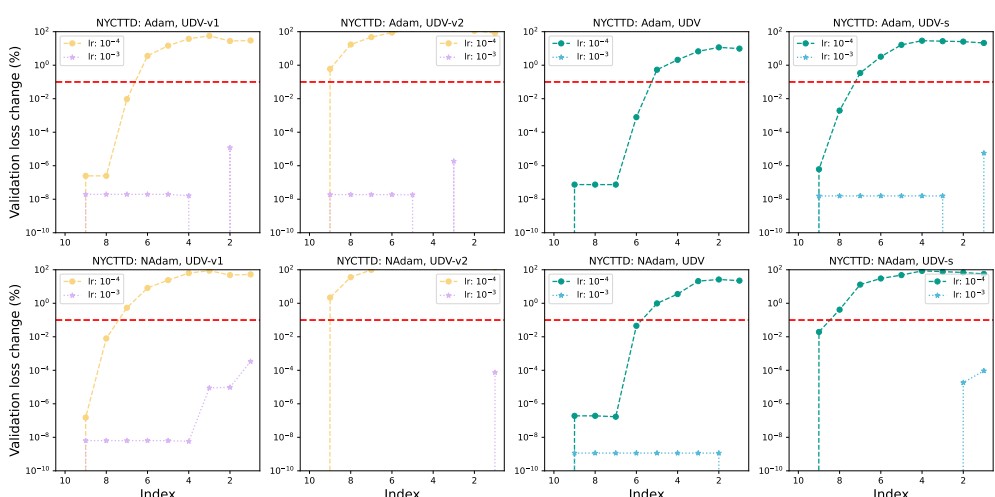

Figure SM15: Different learning rates lead different pruning performance. the validation loss change indicates how much worse the pruned model performs compared to the baseline (the model before pruning), expressed as a percentage ($\frac{loss_{pruned} - loss_{baseline}}{loss_{baseline}} \times 100\%$). Note that the pruned model may outperform the baseline, but negative values cannot be displayed on a logarithmic scale. The red dashed line denotes the 0.1% threshold, indicating negligible performance sacrifice.

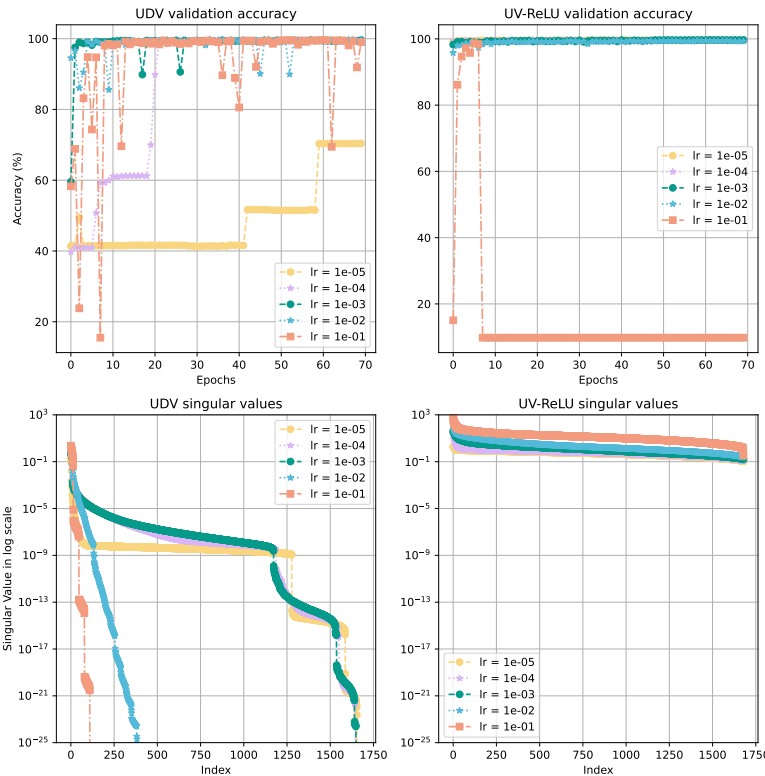

Figure SM16: The "stair-step" accuracy curve in classification (RegNetX-32GF with Adam). The sub-figures in the second row correspond to the respective singular value spectra, indicating that UDV consistently seeks low-rank solutions regardless of the learning rate.

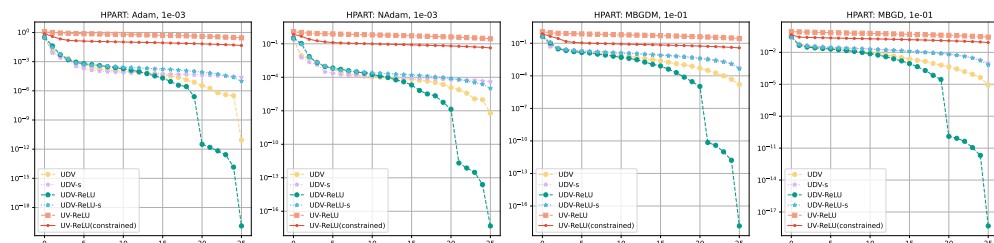

Figure SM17: ReLU in UDV: Singular value pattern on HPART dataset.

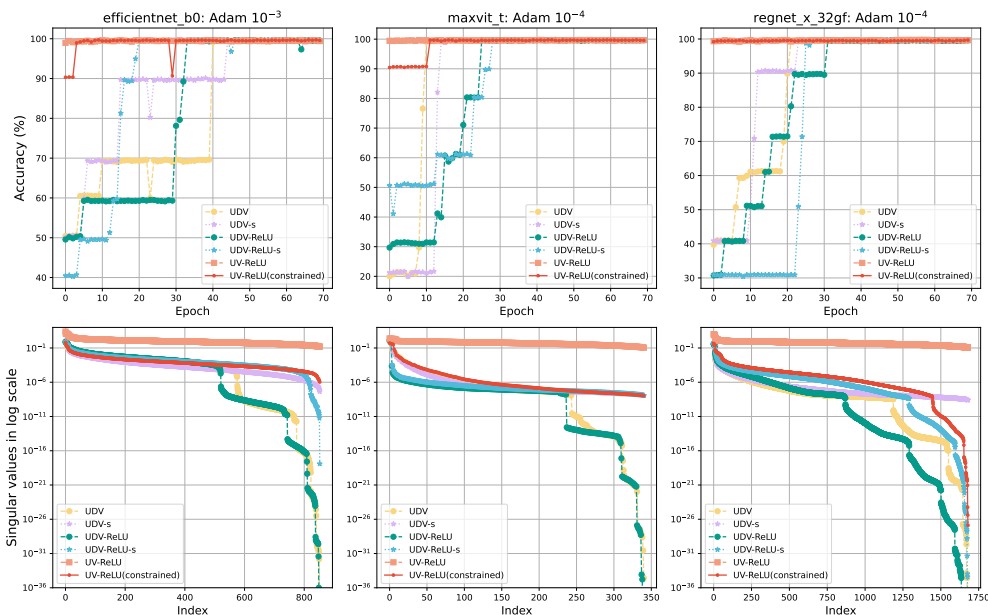

Figure SM18: ReLU in UDV: Validation accuracy (the first row) and the singular value pattern (the second row) on MNIST dataset. Faster convergence can be achieved by fine-tuning the learning rate.

and learning rates from Tables SM2 to SM5, but only used the HPART dataset for regression and the MNIST for classification.

Figure SM17 and Figure SM18 showed that UDV-ReLU and UDV-ReLU-s exhibit a similar (or even more pronounced) decaying pattern in singular values as observed in the proposed UDV. The "stair-step" accuracy curve was also observed in these models.

### B.5 COMPARISON OF UDV AGAINST TWO/THREE-LAYER NETWORK BLOCKS

We extended the experiments to emphasize the necessity of UDV from both structural and constraint perspectives, demonstrating its effectiveness in achieving a low-rank solution. The model we examined included UDV, UDV (unconstrained), UFV and UV. The UDV (unconstrained) model has the same architecture as UDV but without any constraints on the weights, while UFV replaces the diagonal layer of UDV (unconstrained) with a fully connected layer.

The experimental settings followed those outlined in Table 1, and all results are consolidated in Tables SM2 to SM5 Figure SM19 and Figure SM20 demonstrate that the rates of decrease in the singular value spectrum for UDV (unconstrained), UFV and UV are significantly slower compared to UDV. This highlights the importance of the diagonal layer and constraints in enabling the identification of low-rank solutions within the proposed structure.

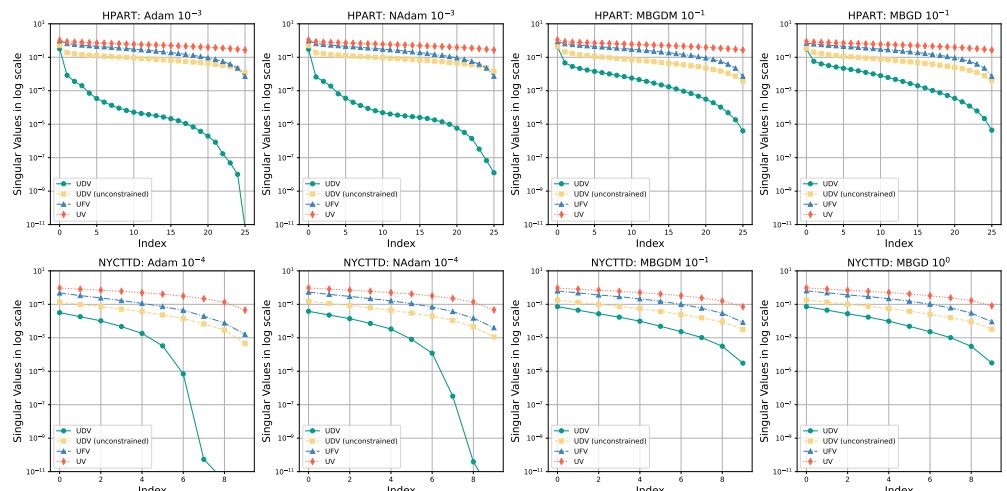

Figure SM19: Comparison of singular value spectrum among UDV, UDV (unconstrained),UFV and UV on the regression datasets.

### B.6  COMPARISON OF UDV AGAINST TWO/THREE-LAYER NETWORK BLOCKS WITH WEIGHT DECAY

Weight decay introduces a regularization term, typically the $L_2$ penalty, into the loss function. This penalizes large weight values and encourages the weights to shrink toward smaller magnitudes. By pushing smaller singular values closer to zero, weight decay tends to lower the rank of the weight matrix. To investigate this effect, we conducted a preliminary experiment to compare the singular value decay observed in two-layer (UV) and three-layer (UFV) fully connected network blocks with weight decay regularization.

We followed the experimental settings outlined in Table 1, focusing on a regression task with the HPART dataset. The weight decay rate ($\gamma$) was incorporated as a regularization parameter in the optimizer, and we evaluated its effect using values $\gamma \in [10^{-6}, 10^{-5}, \ldots, 10^{-1}]$.

Increasing the regularization parameter led to faster decay in the singular value spectrum, but this came at the cost of worse training and validation losses. Figure SM21 shows an example where we selected the largest values for which the validation loss was at most 10% worse than the UDV baseline. The resulting singular value decay in the UV and UFV models was less pronounced than that of the UDV model. When the regularization parameter was increased further to match or surpass the spectral decay of the UDV model, the resulting validation loss deteriorated significantly compared to the UDV network. The complete results are demonstrated in Figures SM22 and SM23.

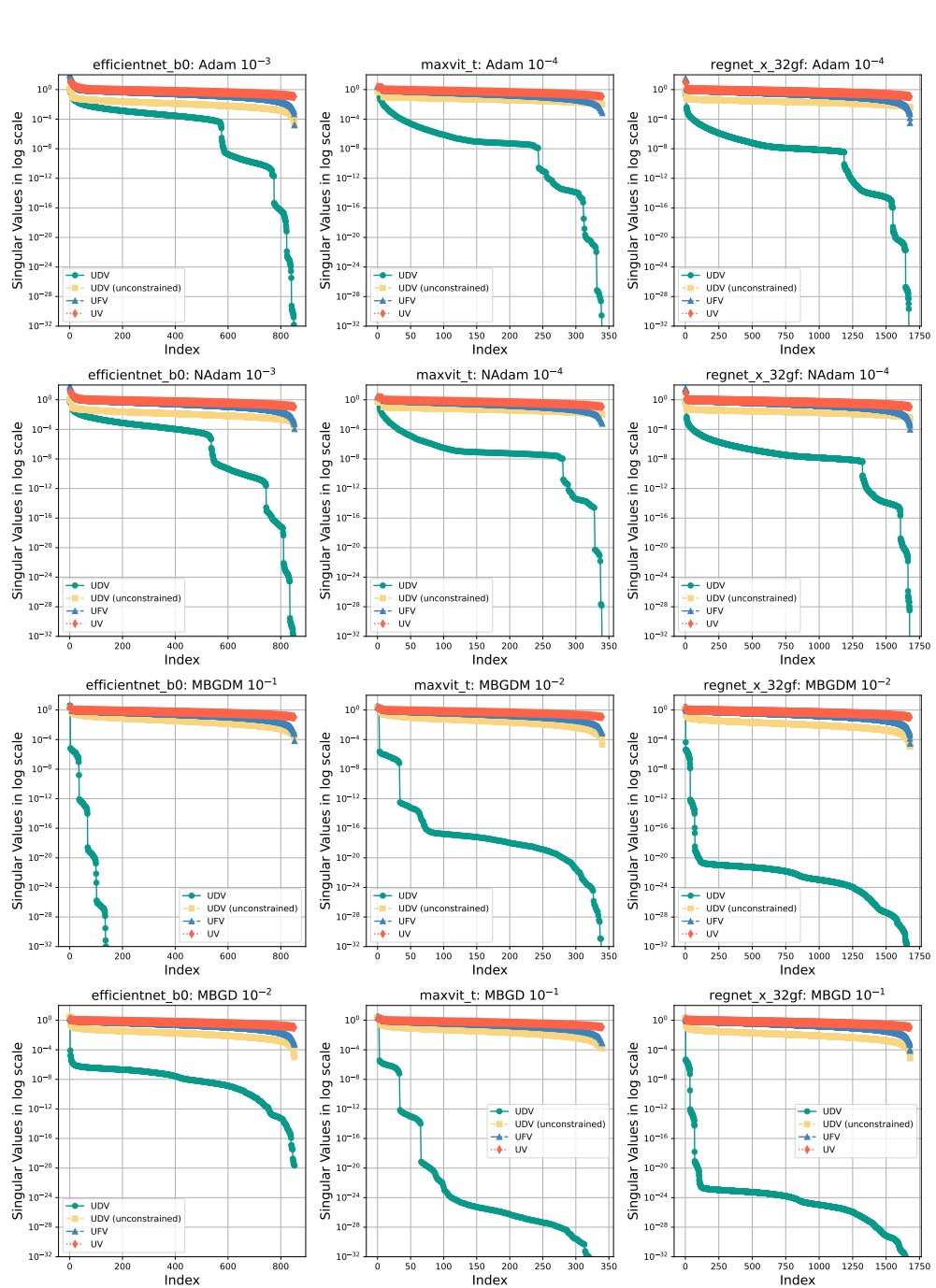

Figure SM20: Comparison of singular value spectrum among UDV, UDV (unconstrained), UFV and UV on the classification datasets (MNIST).

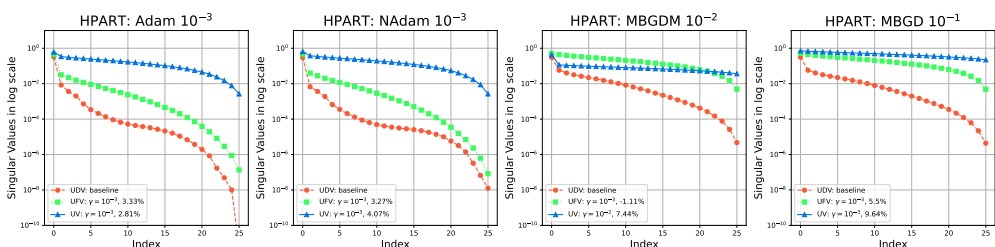

Figure SM21: Comparison of singular value spectrum among the proposed UDV, UFV and UV (with weight decay) on the HPART dataset. In addition to the UDV baseline, the UFV and UV models include the regularization parameter $\gamma$ and the relative change in validation performance. The relative change is defined as ($\frac{loss_{model} - loss_{baseline}}{loss_{baseline}} \times 100\%$), representing how much worse the model performs compared to the UDV baseline.

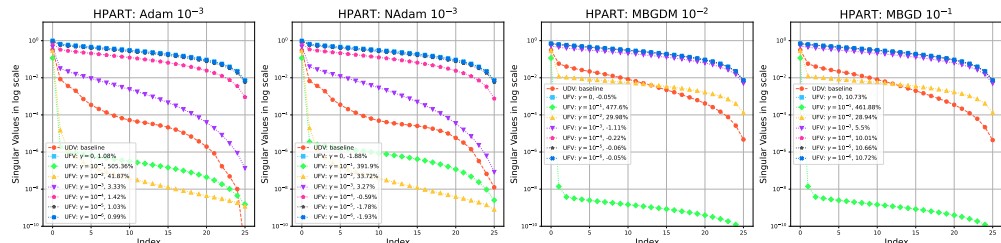

Figure SM22: Extension of Figure SM21. Comparison of singular value spectrum between the proposed UDV and UFV (with weight decay) on the HPART dataset. Regularization parameter $\gamma = 0$ indicates that no weight decay is applied.

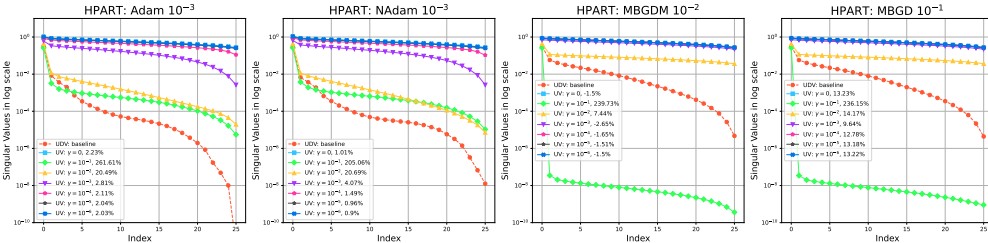

Figure SM23: Extension of Figure SM21. Comparison of singular value spectrum between the proposed UDV and UV (with weight decay) on the HPART dataset. Regularization parameter $\gamma = 0$ indicates that no weight decay is applied.

Table SM2: Experiments using Adam optimizer. Not applicable or results with obvious oscillations or divergence are denoted as '−'.

| Tasks | Regression (Test Loss) | | Classification (Test Accuracy) | | |
|---|---|---|---|---|---|
| Dataset (Transferred model) | HPART − (×10⁻³) | NYCTTD − (×10⁻⁶) | MNIST (MaxVit-T [M] \| EfficientNet-B0 [E] \| RegNetX-32GF [R]) (×100%) | | |
| LR: $10^{-6}$ | − | − | UDV: –
UDV-s: –
UDV-v1: 99.56
UDV-v2: 99.53
UDV-ReLU: –
UDV(unconstrained): 99.53
UFV: 99.53
UV-ReLU: 99.59
UV: 99.54
M: 99.42 | UDV: –
UDV-s: –
UDV-v1: –
UDV-v2: –
UDV-ReLU: –
UDV(unconstrained): –
UFV: 99.25
UV-ReLU: 99.29
UV: 99.24
E: 98.67 | UDV: –
UDV-s: –
UDV-v1: 99.46
UDV-v2: 99.48
UDV-ReLU: –
UDV(unconstrained): 99.48
UFV: 99.38
UV-ReLU: 99.41
UV: 99.36
R: 99.32 |
| LR: $10^{-5}$ | − | − | UDV: –
UDV-s: –
UDV-v1: 99.61
UDV-v2: 99.58
UDV-ReLU: –
UDV(unconstrained): 99.62
UFV: 99.60
UV-ReLU: 99.57
UV: 99.61
M: 99.59 | UDV: –
UDV-s: –
UDV-v1: 99.49
UDV-v2: 99.53
UDV-ReLU: –
UDV(unconstrained): 99.54
UFV: 99.43
UV-ReLU: 99.52
UV: 99.52
E: 99.51 | UDV: –
UDV-s: –
UDV-v1: **99.58**
UDV-v2: 99.50
UDV-ReLU: –
UDV(unconstrained): 99.50
UFV: 99.58
UV-ReLU: 99.59
UV: 99.57
R: **99.59** |
| LR: $10^{-4}$ | UDV: 2.304
UDV-s: 1.912
UDV-v1: 1.823
UDV-v2: 1.738
UDV-ReLU: 2.731
UDV(unconstrained): 1.738
UFV: 1.351
UV-ReLU: 1.376
UV: 1.475 | UDV: **5.248**
UDV-s: 5.248
UDV-v1: **5.248**
UDV-v2: 5.250
UDV-ReLU: 5.251
UDV(unconstrained): 5.250
UFV: 5.254
UV-ReLU: **5.263**
UV: 5.275 | UDV: **99.66**
UDV-s: 99.58
UDV-v1: **99.65**
UDV-v2: **99.69**
UDV-ReLU: **99.63**
UDV(unconstrained): **99.66**
UFV: **99.69**
UV-ReLU: **99.63**
UV: **99.64**
M: **99.65** | UDV: –
UDV-s: –
UDV-v1: **99.63**
UDV-v2: **99.58**
UDV-ReLU: –
UDV(unconstrained): **99.63**
UFV: **99.59**
UV-ReLU: **99.54**
UV: 99.56
E: 99.60 | UDV: **99.55**
UDV-s: **99.55**
UDV-v1: 99.56
UDV-v2: **99.68**
UDV-ReLU: **99.55**
UDV(unconstrained): **99.59**
UFV: **99.67**
UV-ReLU: 99.64
UV: **99.60**
R: 99.59 |
| LR: $10^{-3}$ | UDV: **1.304**
UDV-s: **1.316**
UDV-v1: **1.267**
UDV-v2: **1.268**
UDV-ReLU: **1.320**
UDV(unconstrained): **1.268**
UFV: **1.318**
UV-ReLU: **1.167**
UV: **1.333** | UDV: 5.248
UDV-s: 5.248
UDV-v1: 5.248
UDV-v2: **5.248**
UDV-ReLU: 5.251
UDV(unconstrained): **5.248**
UFV: **5.248**
UV-ReLU: 5.306
UV: **5.251** | UDV: 99.57
UDV-s: 99.59
UDV-v1: 99.57
UDV-v2: 99.58
UDV-ReLU: 99.51
UDV(unconstrained): 99.52
UFV: 99.59
UV-ReLU: 99.60
UV: 99.60
M: 99.63 | UDV: **99.55**
UDV-s: **99.59**
UDV-v1: 99.59
UDV-v2: 99.57
UDV-ReLU: **99.54**
UDV(unconstrained): 99.57
UFV: 99.54
UV-ReLU: 99.49
UV: **99.60**
E: **99.61** | UDV: 99.50
UDV-s: 99.49
UDV-v1: 99.47
UDV-v2: 99.47
UDV-ReLU: 99.42
UDV(unconstrained): 99.58
UFV: 99.41
UV-ReLU: **99.66**
UV: 99.46
R: 99.49 |
| LR: $10^{-2}$ | UDV: 1.877
UDV-s: 1.998
UDV-v1: 1.409
UDV-v2: 1.500
UDV-ReLU: 1.699
UDV(unconstrained): 1.402
UFV: 1.483
UV-ReLU: 1.467
UV: 1.430 | UDV: 5.257
UDV-s: 5.258
UDV-v1: 5.256
UDV-v2: 5.257
UDV-ReLU: 5.323
UDV(unconstrained): 5.263
UFV: 7.048
UV-ReLU: 5.323
UV: 7.369 | UDV: –
UDV-s: –
UDV-v1: –
UDV-v2: –
UDV-ReLU: –
UDV(unconstrained): –
UFV: –
UV-ReLU: –
UV: –
M: – | UDV: 99.34
UDV-s: 99.56
UDV-v1: 99.48
UDV-v2: 99.44
UDV-ReLU: 96.95
UDV(unconstrained): 99.52
UFV: –
UV-ReLU: 99.37
UV: 99.32
E: 98.42 | UDV: 99.53
UDV-s: 99.43
UDV-v1: 99.35
UDV-v2: 99.28
UDV-ReLU: 99.37
UDV(unconstrained): 99.10
UFV: –
UV-ReLU: 99.32
UV: 98.90
R: 99.39 |
| LR: $10^{-1}$ | UDV: 4.188
UDV-s: –
UDV-v1: –
UDV-v2: –
UDV-ReLU: 18.26
UDV(unconstrained): –
UFV: 1.745
UV-ReLU: 42.01
UV: 1.614 | UDV: 5.321
UDV-s: 114.6
UDV-v1: 23.12
UDV-v2: 21.22
UDV-ReLU: 5.323
UDV(unconstrained): 126.1
UFV: –
UV-ReLU: 5.323
UV: – | UDV: –
UDV-s: –
UDV-v1: –
UDV-v2: –
UDV-ReLU: –
UDV(unconstrained): –
UFV: –
UV-ReLU: –
UV: –
M: – | UDV: 99.05
UDV-s: 95.62
UDV-v1: –
UDV-v2: –
UDV-ReLU: –
UDV(unconstrained): –
UFV: –
UV-ReLU: –
UV: 97.69
E: 99.10 | UDV: 97.54
UDV-s: 99.31
UDV-v1: –
UDV-v2: –
UDV-ReLU: –
UDV(unconstrained): –
UFV: –
UV-ReLU: –
UV: –
R: 99.22 |
| LR: $10^{0}$ | UDV: 38.71
UDV-s: –
UDV-v1: 2.413
UDV-v2: 4.633
UDV-ReLU: 48.62
UDV(unconstrained): 14.05
UFV: –
UV-ReLU: 48.24
UV: – | UDV: 5.327
UDV-s: –
UDV-v1: 16.19
UDV-v2: –
UDV-ReLU: 5.323
UDV(unconstrained): –
UFV: –
UV-ReLU: 5.323
UV: – | UDV: –
UDV-s: –
UDV-v1: –
UDV-v2: –
UDV-ReLU: –
UDV(unconstrained): –
UFV: –
UV-ReLU: –
UV: –
M: – | UDV: –
UDV-s: –
UDV-v1: –
UDV-v2: –UDV-ReLU: –
UDV(unconstrained): –
UFV: –
UV-ReLU: –
UV: –
E: – | UDV: –
UDV-s: –
UDV-v1: –
UDV-v2: –
UDV-ReLU: –
UDV(unconstrained): –
UFV: –
UV-ReLU: –
UV: –
R: 95.65 |
| LR: $2 \times 10^{0}$ | UDV: 60.46
UDV-s: –
UDV-v1: 6.651
UDV-v2: 7.366
UDV-ReLU: 48.62
UDV(unconstrained): –
UFV: –
UV-ReLU: 49.43
UV: – | UDV: 5.322
UDV-s: –
UDV-v1: –
UDV-v2: –
UDV-ReLU: 5.323
UDV(unconstrained): –
UFV: –
UV-ReLU: 5.323
UV: – | − | − | − |
| LR: $3 \times 10^{0}$ | UDV: 106.9
UDV-s: –
UDV-v1: 6.606
UDV-v2: 7.398
UDV-ReLU: 48.62
UDV(unconstrained): –
UFV: –
UV-ReLU: 48.50
UV: – | UDV: 5.335
UDV-s: –
UDV-v1: 6.380
UDV-v2: –
UDV-ReLU: 5.323
UDV(unconstrained): –
UFV: –
UV-ReLU: 5.323
UV: – | − | − | − |
| LR: $5 \times 10^{0}$ | − | − | − | − | − |

Table SM3: Experiments using NAdam optimizer. Not applicable or results with obvious oscillations or divergence are denoted as '−'.

| Tasks | Regression (Test Loss) | | Classification (Test Accuracy) | | |
|---|---|---|---|---|---|
| Dataset (Transferred model) | HPART −  ($\times 10^{-3}$) | NYCTTD −  ($\times 10^{-6}$) | MNIST (MaxVit-T [M] \| EfficientNet-B0 [E] \| RegNetX-32GF [R]) ($\times 100\%$) | | |
| LR: $10^{-6}$ | – | – | UDV: –
UDV-s: –
UDV-v1: 99.56
UDV-v2: 99.53
UDV-ReLU: –
UDV(unconstrained): 99.53
UFV: 99.54
UV-ReLU: 99.59
UV: 99.53
M: 99.41 | UDV: –
UDV-s: –
UDV-v1: –
UDV-v2: –
UDV-ReLU: –
UDV(unconstrained): –
UFV: 99.26
UV-ReLU: 99.30
UV: 99.24
E: 98.67 | UDV: –
UDV-s: –
UDV-v1: 99.49
UDV-v2: 99.46
UDV-ReLU: –
UDV(unconstrained): 99.46
UFV: 99.32
UV-ReLU: 99.40
UV: 99.32
R: 99.37 |
| LR: $10^{-5}$ | – | – | UDV: –
UDV-s: –
UDV-v1: 99.63
UDV-v2: 99.59
UDV-ReLU: –
UDV(unconstrained): 99.60
UFV: 99.62
UV-ReLU: 99.61
UV: 99.63
**M: 99.62** | UDV: –
UDV-s: –
UDV-v1: 99.50
UDV-v2: 99.52
UDV-ReLU: –
UDV(unconstrained): 99.52
UFV: 99.43
UV-ReLU: 99.51
UV: 99.52
E: 99.50 | UDV: –
UDV-s: –
UDV-v1: 99.66
UDV-v2: 99.56
UDV-ReLU: –
UDV(unconstrained): 99.56
UFV: 99.54
UV-ReLU: 99.59
UV: 99.61
**R: 99.67** |
| LR: $10^{-4}$ | UDV: 2.312
UDV-s: 1.916
UDV-v1: 1.837
UDV-v2: 1.752
UDV-ReLU: 2.665
UDV(unconstrained): 1.752
**UFV: 1.381**
UV-ReLU: 1.398
UV: 1.512 | UDV: **5.248**
UDV-s: 5.248
UDV-v1: 5.248
UDV-v2: 5.249
UDV-ReLU: 5.251
UDV(unconstrained): 5.249
UFV: 5.256
UV-ReLU: 5.258
UV: 5.275 | UDV: 99.67
UDV-s: **99.62**
UDV-v1: **99.67**
UDV-v2: **99.61**
UDV-ReLU: **99.63**
UDV(unconstrained): 99.62
UFV: **99.68**
UV-ReLU: **99.68**
UV: **99.65**
M: 99.59 | UDV: –
UDV-s: –
UDV-v1: 99.60
UDV-v2: **99.66**
UDV-ReLU: –
UDV(unconstrained): **99.60**
UFV: **99.54**
UV-ReLU: 99.53
UV: 99.53
E: **99.65** | UDV: **99.52**
UDV-s: **99.55**
UDV-v1: **99.69**
UDV-v2: **99.72**
UDV-ReLU: 99.46
UDV(unconstrained): **99.63**
UFV: **99.55**
UV-ReLU: **99.63**
UV: **99.64**
R: 99.64 |
| LR: $10^{-3}$ | **UDV: 1.638**
**UDV-s: 1.691**
**UDV-v1: 1.418**
**UDV-v2: 1.440**
**UDV-ReLU: 1.504**
**UDV(unconstrained): 1.437**
UFV: 1.607
**UV-ReLU: 1.367**
UV: 1.654 | UDV: 5.248
UDV-s: 5.248
UDV-v1: 5.248
UDV-v2: **5.248**
UDV-ReLU: 5.251
UDV(unconstrained): **5.248**
UFV: 5.248
UV-ReLU: 5.249
UV: **5.254** | UDV: 99.53
UDV-s: 99.56
UDV-v1: 99.54
UDV-v2: 99.61
UDV-ReLU: 99.55
UDV(unconstrained): **99.66**
UFV: 99.57
UV-ReLU: 99.63
UV: 99.59
M: 99.58 | UDV: 97.56
UDV-s: **99.61**
UDV-v1: **99.60**
UDV-v2: 99.55
UDV-ReLU: **99.50**
UDV(unconstrained): 99.55
UFV: 99.28
UV-ReLU: **99.55**
UV: **99.55**
E: 99.58 | UDV: 99.45
UDV-s: 99.45
UDV-v1: 99.43
UDV-v2: 99.46
UDV-ReLU: 99.47
UDV(unconstrained): 99.50
UFV: 99.27
UV-ReLU: 99.53
UV:99.42
R: 99.65 |
| LR: $10^{-2}$ | UDV: 3.396
UDV-s: 3.287
UDV-v1: 2.297
UDV-v2: 2.315
UDV-ReLU: 2.403
UDV(unconstrained): 1.884
UFV: –
**UV-ReLU: 1.954**
UV: – | UDV: 5.258
UDV-s: 6.918
UDV-v1: 5.276
UDV-v2: 5.253
UDV-ReLU: 5.323
UDV(unconstrained): 5.253
UFV: 25.24
UV-ReLU: 5.323
UV: 6.262 | UDV: –
UDV-s: –
UDV-v1: 99.31
UDV-v2: –
UDV-ReLU: –
UDV(unconstrained): –
UFV: –
UV-ReLU: –
UV: –
M: – | UDV: **99.35**
UDV-s: –
UDV-v1: 99.31
UDV-v2: 99.01
UDV-ReLU: 99.42
UDV(unconstrained): 98.78
UFV: –
UV-ReLU: 98.87
UV: 98.60
E: 99.26 | UDV: 97.12
UDV-s: 99.28
UDV-v1: 99.37
UDV-v2: 99.38
UDV-ReLU: **99.49**
UDV(unconstrained): 99.38
UFV: –
UV-ReLU: 99.28
UV: 98.91
R: 99.59 |
| LR: $10^{-1}$ | UDV: 13.69
UDV-s: –
UDV-v1: –
UDV-v2: –
UDV-ReLU: 48.62
UDV(unconstrained): –
UFV: 4.918
UV-ReLU: 41.61
**UV: 1.863** | UDV: 5.323
UDV-s: –
UDV-v1: 6.627
UDV-v2: –
UDV-ReLU: 5.323
UDV(unconstrained): –
UFV: –
UV-ReLU: 5.323
UV: – | UDV: –
UDV-s: –
UDV-v1: –
UDV-v2: –
UDV-ReLU: –
UDV(unconstrained): –
UFV: –
UV-ReLU: –
UV: –
M: – | UDV: –
UDV-s: –
UDV-v1: –
UDV-v2: –
UDV-ReLU: –
UDV(unconstrained): –
UFV: –
UV-ReLU: –
UV: 97.54
E: 98.75 | UDV: –
UDV-s: –
UDV-v1: –
UDV-v2: –
UDV-ReLU: –
UDV(unconstrained): –
UFV: –
UV-ReLU: –
UV: –
R: 99.19 |
| LR: $10^{0}$ | UDV: –
UDV-s: –
UDV-v1: 5.649
**UDV-v2: 2.317**
UDV-ReLU: 48.62
UDV(unconstrained): –
UFV: –
UV-ReLU: 45.77
UV: – | UDV: 5.328
UDV-s: 53.70
UDV-v1: 19.52
UDV-v2: –
UDV-ReLU: 5.323
UDV(unconstrained): –
UFV: –
UV-ReLU: 5.323
UV: – | UDV: –
UDV-s: –
UDV-v1: –
UDV-v2: –
UDV-ReLU: –
UDV(unconstrained): –
UFV: –
UV-ReLU: –
UV: –
M: – | UDV: –
UDV-s: –
UDV-v1: –
UDV-v2: –
UDV-ReLU: –
UDV(unconstrained): –
UFV: –
UV-ReLU: –
UV: –
E: – | UDV: –
UDV-s: –
UDV-v1: –
UDV-v2: –
UDV-ReLU: –
UDV(unconstrained): –
UFV: –
UV-ReLU: –
UV: –
R: – |
| LR: $2 \times 10^{0}$ | UDV: –
UDV-s: –
UDV-v1: 3.075
**UDV-v2: 2.828**
UDV-ReLU: 48.62
UDV(unconstrained): –
UFV: –
UV-ReLU: 59.82
UV: – | UDV:5.377
UDV-s: –
UDV-v1: 6.011
UDV-v2: –
UDV-ReLU: 5.323
UDV(unconstrained): –
UFV: –
UV-ReLU: 5.323
UV: – | – | – | – |
| LR: $3 \times 10^{0}$ | UDV: –
UDV-s: –
UDV-v1: 3.882
**UDV-v2: 3.128**
UDV-ReLU: 48.62
UDV(unconstrained): –
UFV: –
UV-ReLU: –
UV: – | UDV: 5.401
UDV-s: –
UDV-v1: 15.27
UDV-v2: –
UDV-ReLU: 5.323
UDV(unconstrained): –
UFV: –
UV-ReLU: 5.323
UV: – | – | – | – |
| LR: $5 \times 10^{0}$ | – | – | – | – | – |

Table SM4: Experiments using MBGD optimizer. Not applicable or results with obvious oscillations or divergence are denoted as '−'.

| Tasks | Regression (Test Loss) | | Classification (Test Accuracy) | | |
|---|---|---|---|---|---|
| Dataset (Transferred model) | HPART −(×10⁻³) | NYCTTD −(×10⁻⁶) | MNIST (MaxVit-T [M] \| EfficientNet-B0 [E] \| RegNetX-32GF [R]) (×100%) | | |
| LR: $10^{-6}$ | − | − | − | − | − |
| LR: $10^{-5}$ | − | − | − | − | − |
| LR: $10^{-4}$ | UDV: 46.90
UDV-s: 45.43
UDV-v1: 46.01
UDV-v2: 44.35
UDV-ReLU: 47.88
UDV(unconstrained): 44.35
UFV: 11.77
UV-ReLU: 12.59
UV: 11.98 | UDV: 26.60
UDV-s: 40.06
UDV-v1: 49.35
UDV-v2: 83.91
UDV-ReLU: 20.56
UDV(unconstrained): 83.91
UFV: 86.68
UV-ReLU: −
UV: 71.72 | − | − | − |
| LR: $10^{-3}$ | UDV: 30.38
UDV-s: 21.39
UDV-v1: 23.65
UDV-v2: 15.77
UDV-ReLU: 39.89
UDV(unconstrained): 15.77
UFV: 6.719
UV-ReLU: 6.478
UV: 6.345 | UDV: 9.407
UDV-s: 10.29
UDV-v1: 10.83
UDV-v2: 12.56
UDV-ReLU: 7.231
UDV(unconstrained): 12.56
UFV: 8.236
UV-ReLU: 12.67
UV: 6.763 | UDV: −
UDV-s: −
UDV-v1: 98.44
UDV-v2: 98.32
UDV-ReLU: −
UDV(unconstrained): 98.27
UFV: 99.34
UV-ReLU: 99.24
UV: 99.19
M: 99.29 | UDV: −
UDV-s: −
UDV-v1: −
UDV-v2: −
UDV-ReLU: −
UDV(unconstrained): −
UFV: 98.15
UV-ReLU: 98.48
UV: 98.55
E: 98.74 | UDV: −
UDV-s: −
UDV-v1: −
UDV-v2: −
UDV-ReLU: −
UDV(unconstrained): −
UFV: 99.20
UV-ReLU: 99.16
UV: 99.17
R: 99.16 |
| LR: $10^{-2}$ | UDV: 6.030
UDV-s: 6.014
UDV-v1: 6.125
UDV-v2: 5.877
UDV-ReLU: 6.234
UDV(unconstrained): 5.877
UFV: 5.575
UV-ReLU: 2.253
UV: 2.251 | UDV: 5.465
UDV-s: 5.421
UDV-v1: 5.514
UDV-v2: 5.532
UDV-ReLU: 5.431
UDV(unconstrained): 5.532
UFV: 5.293
UV-ReLU: 5.521
UV: 5.296 | UDV: 99.38
UDV-s: 99.32
UDV-v1: 99.47
UDV-v2: 99.50
UDV-ReLU: 99.53
UDV(unconstrained): 99.58
UFV: 99.53
UV-ReLU: 99.53
UV: 99.50
M: 99.59 | UDV: −
UDV-s: −
UDV-v1: 98.62
UDV-v2: 98.97
UDV-ReLU: −
UDV(unconstrained): 98.40
UFV: 99.05
UV-ReLU: 99.31
UV: 99.24
E: 99.43 | UDV: 99.40
UDV-s: 99.46
UDV-v1: 99.37
UDV-v2: 99.34
UDV-ReLU: **99.47**
UDV(unconstrained): 99.34
UFV: 99.36
UV-ReLU: 99.31
UV: 99.38
R: 99.44 |
| LR: $10^{-1}$ | UDV: **1.398**
UDV-s: **1.407**
UDV-v1: **1.556**
UDV-v2: **1.565**
UDV-ReLU: **1.379**
UDV(unconstrained): **1.565**
UFV: **1.548**
UV-ReLU: **1.493**
UV: **1.583** | UDV: 5.253
UDV-s: 5.250
UDV-v1: 5.256
UDV-v2: 5.252
UDV-ReLU: 5.277
UDV(unconstrained): 5.252
UFV: **5.255**
UV-ReLU: **5.323**
UV: **5.291** | UDV: **99.59**
UDV-s: **99.38**
UDV-v1: **99.61**
UDV-v2: **99.60**
UDV-ReLU: **99.65**
UDV(unconstrained): **99.59**
UFV: **99.61**
UV-ReLU: **99.60**
UV: **99.63**
M: **99.62** | UDV: −
UDV-s: 99.36
UDV-v1: 99.43
UDV-v2: 99.53
UDV-ReLU: 95.21
UDV(unconstrained): **95.29**
UFV: **99.55**
UV-ReLU: 99.52
UV: 99.55
E: **99.67** | UDV: **99.51**
UDV-s: **99.57**
UDV-v1: 99.60
UDV-v2: 99.62
UDV-ReLU: −
UDV(unconstrained): **99.61**
UFV: 99.59
UV-ReLU: 99.59
UV: 98.56
R: 99.63 |
| LR: $10^0$ | UDV: 3.076
UDV-s: 2.870
UDV-v1: 1.935
UDV-v2: 1.830
UDV-ReLU: 4.573
UDV(unconstrained): 1.810
UFV: −
UV-ReLU: 48.62
UV: − | UDV: **5.248**
UDV-s: **5.248**
UDV-v1: **5.249**
UDV-v2: **5.248**
UDV-ReLU: 5.261
UDV(unconstrained): **5.248**
UFV: 5.256
UV-ReLU: 5.271
UV: − | UDV: −
UDV-s: −
UDV-v1: −
UDV-v2: −
UDV-ReLU: −
UDV(unconstrained): −
UFV: −
UV-ReLU: −
UV: −
M: − | UDV: −
UDV-s: −
UDV-v1: **99.65**
UDV-v2: 99.59
UDV-ReLU: −
UDV(unconstrained): 99.62
UFV: −
UV-ReLU: **99.64**
UV: −
E: 99.55 | UDV: −
UDV-s: −
UDV-v1: **99.67**
UDV-v2: **99.69**
UDV-ReLU: −
UDV(unconstrained): 99.57
UFV: −
UV-ReLU: **99.73**
UV: **99.66**
R: **99.66** |
| LR: $2 \times 10^0$ | UDV: 6.244
UDV-s: 48.63
UDV-v1: 2.559
UDV-v2: −
UDV-ReLU: 47.50
UDV(unconstrained): −
UFV: −
UV-ReLU: 48.63
UV: − | UDV: 5.248
UDV-s: 5.248
UDV-v1: 5.249
UDV-v2: 5.249
UDV-ReLU: 5.259
UDV(unconstrained): 5.249
UFV: −
UV-ReLU: 5.293
UV: − | UDV: −
UDV-s: −
UDV-v1: −
UDV-v2: −
UDV-ReLU: −
UDV(unconstrained): −
UFV: −
UV-ReLU: −
UV: −
M: − | UDV: −
UDV-s: −
UDV-v1: 98.37
UDV-v2: **99.66**
UDV-ReLU: −
UDV(unconstrained): −
UFV: −
UV-ReLU: −
UV: −
E: 98.88 | UDV: −
UDV-s: −
UDV-v1: 99.66
UDV-v2: 99.55
UDV-ReLU: −
UDV(unconstrained): −
UFV: −
UV-ReLU: 99.58
UV: 99.57
R: 99.56 |
| LR: $3 \times 10^0$ | UDV: −
UDV-s: −
UDV-v1: −
UDV-v2: −
UDV-ReLU: 48.62
UDV(unconstrained): −
UFV: −
UV-ReLU: 48.62
UV: − | UDV: 5.248
UDV-s: 5.248
UDV-v1: 5.249
UDV-v2: 5.249
UDV-ReLU: **5.257**
UDV(unconstrained): 5.250
UFV: −
UV-ReLU: 5.296
UV: − | UDV: −
UDV-s: −
UDV-v1: −
UDV-v2: −
UDV-ReLU: −
UDV(unconstrained): −
UFV: −
UV-ReLU: −
UV: −
M: − | UDV: −
UDV-s: −
UDV-v1: 99.07
UDV-v2: −
UDV-ReLU: −
UDV(unconstrained): −
UFV: −
UV-ReLU: −
UV: −
E: 99.26 | UDV: −
UDV-s: −
UDV-v1: 99.48
UDV-v2: 99.65
UDV-ReLU: −
UDV(unconstrained): −
UFV: −
UV-ReLU: 99.57
UV: −
R: 99.57 |
| LR: $5 \times 10^0$ | − | − | UDV: −
UDV-s: −
UDV-v1: −
UDV-v2: −
UDV-ReLU: −
UDV(unconstrained): −
UFV: −
UV-ReLU: −
UV: −
M: − | UDV: −
UDV-s: −
UDV-v1: −
UDV-v2: −
UDV-ReLU: −
UDV(unconstrained): −
UFV: −
UV-ReLU: −
UV: −
E: 99.33 | UDV: −
UDV-s: −
UDV-v1: 99.55
UDV-v2: 99.41
UDV-ReLU: −
UDV(unconstrained): −
UFV: −
UV-ReLU: 99.42
UV: −
R: 99.55 |

Table SM5: Experiments using MBGDM optimizer. Not applicable or results with obvious oscillations or divergence are denoted as '–'.

| Tasks | Regression (Test Loss) | | Classification (Test Accuracy) | | |
|---|---|---|---|---|---|
| Dataset (Transferred model) | HPART (×10⁻³) | NYCTTD (×10⁻⁶) | MNIST (MaxVit-T [M]) (×100%) | MNIST (EfficientNet-B0 [E]) (×100%) | MNIST (RegNetX-32GF [R]) (×100%) |
| LR: $10^{-6}$ | – | – | – | – | – |
| LR: $10^{-5}$ | – | – | – | – | – |
| LR: $10^{-4}$ | UDV: 30.52
UDV-s: 21.50
UDV-v1: 23.79
UDV-v2: 15.84
UDV-ReLU: 40.00
UDV(unconstrained): 15.84
UFV: 6.729
UV-ReLU: 6.488
UV: 6.348 | UDV: 9.413
UDV-s: 10.29
UDV-v1: 10.85
UDV-v2: 12.59
UDV-ReLU: 7.224
UDV(unconstrained): 12.59
UFV: 8.288
UV-ReLU: 12.68
UV: 6.776 | – | – | – |
| LR: $10^{-3}$ | UDV: 6.105
UDV-s: 6.080
UDV-v1: 6.178
UDV-v2: 5.929
UDV-ReLU: 6.341
UDV(unconstrained): 5.929
UFV: 2.590
UV-ReLU: 2.529
UV: 2.252 | UDV: 5.457
UDV-s: 5.412
UDV-v1: 5.516
UDV-v2: 5.534
UDV-ReLU: 5.431
UDV(unconstrained): 5.534
UFV: 5.294
UV-ReLU: 5.520
UV: 5.296 | UDV: 99.26
UDV-s: 99.38
UDV-v1: 99.39
UDV-v2: 99.52
UDV-ReLU: –
UDV(unconstrained): 99.51
UFV: 99.56
UV-ReLU: 99.46
UV: 99.54
M: 99.58 | UDV: –
UDV-s: –
UDV-v1: 99.00
UDV-v2: 99.05
UDV-ReLU: –
UDV(unconstrained): 99.08
UFV: 99.18
UV-ReLU: 99.37
UV: 99.36
E: 99.45 | UDV: 99.50
UDV-s: 99.43
UDV-v1: 99.39
UDV-v2: 99.37
UDV-ReLU: –
UDV(unconstrained): 99.40
UFV: 99.38
UV-ReLU: 99.36
UV: 99.30
R: 99.41 |
| LR: $10^{-2}$ | UDV: 1.357
UDV-s: 1.388
UDV-v1: 1.554
UDV-v2: 1.569
UDV-ReLU: **1.312**
UDV(unconstrained): 1.569
UFV: 1.357
UV-ReLU: 1.314
UV: **1.337** | UDV: 5.253
UDV-s: 5.250
UDV-v1: 5.256
UDV-v2: 5.252
UDV-ReLU: 5.276
UDV(unconstrained): 5.252
UFV: 5.254
UV-ReLU: 5.267
UV: 5.279 | UDV: **99.67**
UDV-s: 99.38
UDV-v1: 99.63
UDV-v2: 99.60
UDV-ReLU: **99.62**
UDV(unconstrained): 99.59
UFV: **99.65**
UV-ReLU: 99.63
UV: **99.69**
M: **99.64** | UDV: 99.48
UDV-s: 99.60
UDV-v1: 99.54
UDV-v2: 99.59
UDV-ReLU: 99.54
UDV(unconstrained): 99.55
UFV: 99.50
UV-ReLU: 99.53
UV: 99.50
E: 99.59 | UDV: **99.74**
UDV-s: **99.72**
UDV-v1: 99.61
UDV-v2: **99.67**
UDV-ReLU: **99.65**
UDV(unconstrained): 99.63
UFV: **99.66**
UV-ReLU: 99.61
UV: 99.53
R: 99.60 |
| LR: $10^{-1}$ | UDV: **1.345**
UDV-s: **1.339**
UDV-v1: **1.313**
UDV-v2: **1.302**
UDV-ReLU: 1.318
UDV(unconstrained): **1.302**
UFV: 1.358
UV-ReLU: **1.244**
UV: – | UDV: **5.248**
UDV-s: 5.248
UDV-v1: 5.249
UDV-v2: 5.248
UDV-ReLU: 5.259
UDV(unconstrained): 2.248
UFV: **5.251**
UV-ReLU: **5.264**
UV: **5.259** | UDV: 99.61
UDV-s: **99.60**
UDV-v1: **99.65**
UDV-v2: **99.68**
UDV-ReLU: –
UDV(unconstrained): **99.70**
UFV: –
UV-ReLU: 99.63
UV: –
M: – | UDV: 99.63
UDV-s: **99.63**
UDV-v1: **99.64**
UDV-v2: **99.66**
UDV-ReLU: **99.61**
UDV(unconstrained): 99.59
UFV: **99.68**
UV-ReLU: **99.68**
UV: **99.59**
E: **99.63** | UDV: 99.60
UDV-s: 99.57
UDV-v1: **99.71**
UDV-v2: 99.67
UDV-ReLU: –
UDV(unconstrained): **99.67**
UFV: 99.63
UV-ReLU: **99.66**
UV: **99.56**
R: **99.67** |
| LR: $10^{0}$ | UDV: 123.6
UDV-s: –
UDV-v1: –
UDV-v2: –
UDV-ReLU: 47.03
UDV(unconstrained): –
UFV: –
UV-ReLU: 48.62
UV: – | UDV: 5.248
UDV-s: 5.248
UDV-v1: 5.249
UDV-v2: 5.249
UDV-ReLU: 5.251
UDV(unconstrained): 5.249
UFV: –
UV-ReLU: 5.281
UV: – | UDV: –
UDV-s: –
UDV-v1: 99.54
UDV-v2: –
UDV-ReLU: –
UDV(unconstrained): –
UFV: –
UV-ReLU: –
UV: –
M: – | UDV: –
UDV-s: –
UDV-v1: 99.54
UDV-v2: –
UDV-ReLU: –
UDV(unconstrained): –
UFV: –
UV-ReLU: –
UV: –
E: 99.24 | UDV: –
UDV-s: –
UDV-v1: 99.52
UDV-v2: 99.58
UDV-ReLU: –
UDV(unconstrained): –
UFV: –
UV-ReLU: 99.53
UV: –
R: 99.34 |
| LR: $2 \times 10^{0}$ | UDV: –
UDV-s: –
UDV-v1: –
UDV-v2: –
UDV-ReLU: 48.62
UDV(unconstrained): –
UFV: –
UV-ReLU: 48.62
UV: – | UDV: 5.248
UDV-s: 5.248
UDV-v1: 5.249
UDV-v2: 5.248
UDV-ReLU: **5.249**
UDV(unconstrained): 5.248
UFV: –
UV-ReLU: 5.289
UV: – | UDV: –
UDV-s: –
UDV-v1: –
UDV-v2: –
UDV-ReLU: –
UDV(unconstrained): –
UFV: –
UV-ReLU: –
UV: –
M: – | UDV: –
UDV-s: –
UDV-v1: –
UDV-v2: –
UDV-ReLU: –
UDV(unconstrained): –
UFV: –
UV-ReLU: –
UV: –
E: 98.95 | UDV: –
UDV-s: –
UDV-v1: 99.31
UDV-v2: 99.37
UDV-ReLU: –
UDV(unconstrained): –
UFV: –
UV-ReLU: 99.28
UV: –
R: 99.31 |
| LR: $3 \times 10^{0}$ | UDV: –
UDV-s: –
UDV-v1: –
UDV-v2: –
UDV-ReLU: 48.62
UDV(unconstrained): –
UFV: –
UV-ReLU: 48.62
UV: – | UDV: 5.248
UDV-s: **5.248**
UDV-v1: **5.248**
UDV-v2: **5.248**
UDV-ReLU: 5.249
UDV(unconstrained): **5.248**
UFV: –
UV-ReLU: 5.292
UV: – | UDV: –
UDV-s: –
UDV-v1: –
UDV-v2: –
UDV-ReLU: –
UDV(unconstrained): –
UFV: –
UV-ReLU: –
UV: –
M: – | UDV: –
UDV-s: –
UDV-v1: –
UDV-v2: –
UDV-ReLU: –
UDV(unconstrained): –
UFV: –
UV-ReLU: –
UV: –
E: 98.70 | UDV: –
UDV-s: –
UDV-v1: –
UDV-v2: –
UDV-ReLU: –
UDV(unconstrained): –
UFV: –
UV-ReLU: –
UV: –
R: 99.12 |
| LR: $5 \times 10^{0}$ | – | – | UDV: –
UDV-s: –
UDV-v1: –
UDV-v2: –
UDV-ReLU: –
UDV(unconstrained): –
UFV: –
UV-ReLU: –
UV: –
M: – | UDV: –
UDV-s: –
UDV-v1: –
UDV-v2: –
UDV-ReLU: –
UDV(unconstrained): –
UFV: –
UV-ReLU: –
UV: –
E: – | UDV: –
UDV-s: –
UDV-v1: 99.55
UDV-v2: 99.41
UDV-ReLU: –
UDV(unconstrained): –
UFV: –
UV-ReLU: 99.42
UV: –
R: 99.55 |

