# OpenReview forum: "Implicit Bias in Matrix Factorization and its Explicit Realization in a new Architecture"
_ICLR.cc/2025/Conference — Submitted to ICLR 2025_

### Official Review · Reviewer_SBgy · 2024-10-24

**Soundness:** 3
**Presentation:** 2
**Contribution:** 2
**Rating:** 3
**Confidence:** 4

**Summary:**

Motivated by the literature on the implicit bias towards low rank in matrix factorization, proposes the $UDV^\top$ (and symmetric $UDU^\top$) factorization, which consists of two norm constrained matrices and a diagonal component. Then, empirically shows that this parameterization enjoys a stronger implicit bias towards low-rank solutions, with a weaker dependence on the learning rate and initialization size. Based on the results for matrix factorization, the proposed factorization is evaluated in the context of transfer learning, where it is used to parameterize the linear head of a non-linear neural network base.

**Strengths:**

For standard matrix factorization, the implicit bias towards low rank usually requires a small initialization. Since the origin is a saddle point, this leads to a tradeoff between optimization time and generalization. I find it interesting that the proposed $UDV^\top$ factorization is demonstrated empirically to enjoy an implicit bias towards low rank even for relatively large initializations (and learning rates). This observation can motivate theoretically studying properties of the $UDV^\top$ factorization.

**Weaknesses:**

1. The paper does not provide insight, whether theoretical or empirical, as to why the proposed matrix factorization enjoys a stronger low rank bias that depends less on learning rate and initialization, compared to other matrix factorizations.

2. Given the above, the significance of the contributions hinges on the applicability of the factorization and its empirical evaluation. However, the empirical evaluation is limited. Specifically, the considered datasets are non-standard and outdated (MNIST and a couple of tabular regression datasets), and the results do not seem to suggest that the proposed $UDV^\top$ factorization brings any advantage compared to a standard matrix factorization or a simple two-layer ReLU network head, in the considered transfer learning setting.

3. The novelty of reducing network size with SVD-based pruning is limited. Prior work has already suggested compressing neural networks through low rank matrix (and tensor) decompositions (e.g. [1,2,3]). Furthermore, there is no comparison to these or other alternative network pruning and compression techniques, and the utility of the method proposed in this paper is unclear since it only pertains to the last linear classifier head, leaving the base of the network uncompressed.

At its current form, I find that this work does not meet the bar required for publication at ICLR. I would advise the authors to strengthen either the empirical or theoretical components of their work. In particular, it would be beneficial to use more standard benchmarks (e.g. CIFAR10, ImageNet, or standard datasets from other domains) and compare the low rank pruning method to existing compression and pruning baselines. In terms of theory, any insight as to why the implicit low rank bias of the $UDV^\top$ factorization is more robust compared to that in standard matrix factorizations would also substantially increase the significance of the contributions.


Additional comments:

- It would be interesting to report in the matrix factorization experiments how the singular values evolve under the proposed $UDU^\top$ factorization. Since $D$ is initialized to be the identity, do some of its diagonal entries shrink to zero? This may be an interesting point to try and explain theoretically.

- In Figure 1, it would be useful to report reconstruction error (i.e. how well the learned solution generalizes) alongside the training loss.

- Isn't Equation (6) just $VDU^\top \mathbf{x}$ in a different notation? I believe the connection to the proposed matrix factorization would be more straightforward by writing the equation in matrix notation. Furthermore, what is the purpose of $ \approx y$ in that equation? I would suggest dropping it.

[1] Li, J., Sun, Y., Su, J., Suzuki, T., & Huang, F. (2020, June). Understanding generalization in deep learning via tensor methods. In International Conference on Artificial Intelligence and Statistics (pp. 504-515). PMLR.

[2] Kim, Y. D., Park, E., Yoo, S., Choi, T., Yang, L., & Shin, D. (2015). Compression of deep convolutional neural networks for fast and low power mobile applications. arXiv preprint arXiv:1511.06530.

[3] Yaguchi, A., Suzuki, T., Nitta, S., Sakata, Y., & Tanizawa, A. (2019). Decomposable-Net: Scalable low-rank compression for neural networks. arXiv preprint arXiv:1910.13141.

**Questions:**

--

---

> ### Author Response · Authors · 2024-11-25
>
> **Lack of insight**
>
> We take this opportunity to highlight the key distinction between achieving approximately low-rank solutions and converging to truly low-rank solutions, which underpins the **novelty, importance, and extraordinary nature of our observations**. Prior theoretical work primarily focuses on achieving approximately low-rank solutions, a goal that is theoretically tractable under certain settings and can often be achieved through standard convex optimization formulations, such as those using nuclear norm penalties. In contrast, our empirical results reveal the remarkable and unexpected observation that the proposed framework consistently converges to **truly low-rank solutions**. This behavior is fundamentally different and cannot be explained by existing theories on approximate low-rank solutions. While providing theoretical guarantees for convergence to strictly low-rank solutions is beyond the current scope of our work, our findings open intriguing directions for future theoretical research.
>
> **Limited Empirical Evaluation**
>
> We respectfully disagree with the reviewer’s assessment that the experiments are limited in scope. Observing the bias toward truly low-rank solutions requires running the models for extended periods, which is particularly evident in our matrix factorization examples. This constraint naturally limits the scale of datasets we can investigate in the current work. Nonetheless, we have designed our experiments to test this phenomenon across diverse settings, including matrix factorization problem, phase retrieval, neural network training with regression, classification, and transfer learning tasks, with multiple datasets and loss functions. These varied scenarios provide strong evidence of the generality and robustness of our findings. We do, however, plan to explore this framework in the context of self-attention and other architectures in future work.
>
> **Pruning**
>
> The main contribution of our paper is the extraordinary observation that UDV networks can identify truly low-rank solutions. We believe this observation is of significant interest to researchers studying the theory of neural networks, as the dynamical behavior exhibited by UDV networks provides insights into the underlying mechanisms of implicit bias in fundamentally different ways compared to existing theories based on gradient flow.
>
> The pruning experiments in our paper are intended as a proof of concept, demonstrating that truly low-rank solutions can also be leveraged for practical purposes. Our goal in this section is not to provide an exhaustive analysis of pruning techniques but rather to illustrate one potential application of our framework. We acknowledge the extensive literature on pruning via low-rank factorization and note that UDV networks can be seamlessly integrated with existing compression and pruning methods.
> We will add the mentioned references to the discussion of related work.

---

> ### Author Response · Authors · 2024-11-25
>
> **Additional comments**
>
> [Does diagonal entries shrink to zero] Yes, they do. The obtained factors are rank-revealing: either the diagonal entries of D or the columns of U converge to zero, leading to truly low-rank solutions. Specifically, U exhibits a tendency to grow along certain directions, while the rescaling induced by projection onto the bounded constraint shrinks other coordinates. This behavior aligns with the mechanism of the power method and provides insights into the connection with divergent forces.
>
> [Add reconstruction error in Figure 1] It is a good idea. We will incorporate this in the revised version. While this requires running the experiment again, we can report the final test accuracies immediately from our saved results.
>
> Below, we report the relative distance between the true matrix ($X^{\natural}$) and the matrix recovered $(X)$ using $UU^T$ and $UDU^T$ methods. The relative error is computed as:
> $$ rel.err = ||X^{\natural} - X||_F / || X^{\natural}||_F $$
>
> For the settings in Fig 1., we get
>
> | Step Size ($\eta$)   | $UU^\top$         | $UDU^\top$             |
> |-----------------------|-------------------|-------------------------|
> | $10^{-3}$            | 0.0220           | 0.0088                 |
> | $10^{-2}$            | 0.0173           | $2.57 \cdot 10^{-9}$   |
> | $10^{-1}$            | 0.0167           | $2.58 \cdot 10^{-10}$  |
>
> | Distance to Origin ($\xi$) | $UU^\top$         | $UDU^\top$             |
> |----------------------------|-------------------|-------------------------|
> | $10^{-4}$                  | 0.0027           | $2.58 \cdot 10^{-9}$   |
> | $10^{-3}$                  | 0.0060           | $2.57 \cdot 10^{-9}$   |
> | $10^{-2}$                  | 0.0173           | $2.58 \cdot 10^{-9}$   |
> | $10^{-1}$                  | 0.0766           | $2.59 \cdot 10^{-9}$   |
> | $10^{0}$                   | 0.3477           | $2.57 \cdot 10^{-9}$   |
>
> For the noisy setting in Appendix A1, Figure SM1, we obtain
>
> | Step Size ($\eta$)   | $UU^\top$   | $UDU^\top$   |
> |-----------------------|-------------|--------------|
> | $10^{-3}$            | 0.4248      | 0.2322       |
> | $10^{-2}$            | 0.6482      | 0.4337       |
> | $10^{-1}$            | 0.6247      | 0.4238       |
>
> | Distance to Origin ($\xi$) | $UU^\top$   | $UDU^\top$   |
> |----------------------------|-------------|--------------|
> | $10^{-4}$                  | 0.7003      | 0.4372       |
> | $10^{-3}$                  | 0.6759      | 0.4374       |
> | $10^{-2}$                  | 0.6482      | 0.4337       |
> | $10^{-1}$                  | 0.6247      | 0.4238       |
> | $10^{0}$                   | 0.6696      | 0.4325       |
>
> [Matrix/vector notation] You are correct. We adopted the explicit sum notation in the neural network training section as it is standard in that domain. We will clarify the connection.

---

> > ### Comment · Reviewer_SBgy · 2024-11-25
> >
> > Thank you for the response. I have read it and the other reviews carefully.
> >
> > The concerns raised in my original review remain largely the same. Due to the reasons stated in the review, although I do believe this work has potential, in my opinion it does not meet the bar for publication at ICLR.

---

### Official Review · Reviewer_fryc · 2024-10-29

**Soundness:** 3
**Presentation:** 3
**Contribution:** 2
**Rating:** 5
**Confidence:** 3

**Summary:**

The main contribution of this work is a novel formulation of the matrix factorization, namely the UDV factorization. The authors conduct a series of numerical experiments to reveal that the proposed formulation has a stronger implicit bias towards low-rank solution than the common factorization $X = UV^T$, which then inspires a structure of neural networks that has the potential to be lightweight by pruning.

**Strengths:**

Overall, this paper is well-written and clearly organized. The idea is simple and straightforward as reflected in Eq. (4). The effect of the proposed formulation is also promising: it clearly exhibits a favor of the low-rank solution, e.g., the singular values decay very fast in Fig. 1. The phenomena observed in this paper definitely give rise to many follow-up questions, e.g., why gradient descent exhibits such a strong preference for low-rank solution compared to other forms of factorization.

**Weaknesses:**

My major concerns are in three aspects.

1. The underlying mechanism of the effectiveness of the proposed formulation is completely absent in the current version of this paper. As there is not a clear and compelling motivation for why the formulation proposed in this paper is highly effective for gradient descent to obtain a low-rank solution, I think it is crucial to give a corresponding comprehensive explanation or discussion regarding the effectiveness of the UDV decomposition, including the roles of all the explicit constraints. Even an informal explanation would be helpful for understanding the idea behind the formulation, e.g., why impose these explicit constraints and why they are helpful for finding low-rank solutions?

2. The authors do not carefully discuss the related work Li et al., 2020. Li et al., 2020 clearly showed that gradient flow (GF) prefers low-rank solution by connecting the GF dynamics with Greedy Low Rank Learning (GLRL). In particular, GF implicitly prefers low-rank solution even when $r = d $ for the factorization $X = UU^T, U \in R^{d \times r}$.

    The lack of a comparison between this work and Li et al., 2020 will induce several key questions. Can the theory in Li et al., 2020 be applied to explain results in this paper as UDV decomposition can be viewed as a three-layer GLRL? What will happen in the UDV decomposition if $r = d$? Does it still exhibit a stronger preference of low-rank solution compared to $UU^T$? What will happen if the constraints on the norms are relaxed, e.g., by assuming a large $\alpha$?

3. The comparison between the proposed formulation and other related factorizations is absent. There are already a variety of formulations for matrix factorization. In particular, if the UDV decomposition can still be applied in the case with $r = d$, then what are the differences between the UDV decomposition and other ones, say LDLT decomposition, and what are advantages and disadvantages of UDV? More specifically, when will UDV decomposition be a better choice than other decompositions?

Minor comments:

How does the rank of $X$ affect qualities of the recovered solutions for different factorizations? For example, in Fig. 1, if $r$ varies from 3 to 20 ($r = 3$ in Fig. 1), does UDV decomposition still find better solution?

**Reference**

Li et al., 2020. Towards Resolving the Implicit Bias of Gradient Descent for Matrix Factorization: Greedy Low-Rank Learning.

**Questions:**

1. Why is it crucial to impose explicit constraints on norms and why they are helpful for finding low-rank solutions?

2. Can depth-3 GLRL in Li et al., 2020 explain the proposed formulation in this work? If this is not the case, then what will happen in the UDV decomposition if $r = d$ and does it still exhibits a stronger preference of low-rank solution over $UU^T$?

3. What are the differences between the UDV decomposition and LDLT decomposition?

---

> ### Author Response · Authors · 2024-11-25
>
> **Lack of insight – why constraints are useful?**
>
> We posit that low-rank bias is in fact induced from the divergent dynamics. In matrix factorization experiments, we observe that the factors $UDU^T$ formulation finds are rank-revealing: either the diagonal entries of D or the columns of U converge to zero, leading to truly low-rank solutions. Specifically, U exhibits a tendency to grow along certain directions, while the rescaling induced by projection onto the bounded constraint shrinks other coordinates. This behavior aligns with the mechanisms observed in the power method and provides insights into the connection with divergent forces.
>
> **Comparison with Li et al. – is this depth 3 of their approach**
>
> We will provide a more detailed discussion of this reference in the revised manuscript. However, their results cannot explain our observations, for instance, we have shown that explicit constraints are crucial to obtain truly low-rank solutions in appendix B5. In the original submission, this comparison was limited to the HPART dataset for a regression task. To address the reviewer's concerns and to strengthen the evaluation, we have now extended this section. Specifically, we have included additional datasets, present results also using cross-entropy loss in classification tasks, and explore transfer learning scenarios. We posit that constraints provide a tighter and more interpretable mechanism for promoting low-rank behavior, as they explicitly restrict the space for U and V. This advantage is particularly evident in our experiments.
>
> **Comparison with other factorizations**
>
> There seems to be a misunderstanding here. The notion of matrix factorization differs slightly between numerical linear algebra and optimization.
>
> In numerical linear algebra, the focus is on exact or stable decompositions, such as LU, QR, or SVD, to solve systems of equations, compute eigenvalues, or invert matrices with high accuracy and efficiency. In optimization, the focus shifts to approximating structured matrices, such as low-rank or sparse representations, to fit specific models or solve specific problems. For example, in our paper, we use the $UDU^\top$ factorization for the matrix completion problem, where the matrix is incomplete. Notably, you cannot compute an $LDL^\top$ decomposition for an incomplete matrix.
>
> **UDV decomposition if r=d**
>
> We believe there is a misunderstanding. We are already using full-rank factors in our experiment. Our matrix factorization problems are $n \times n$ dimensional, and we choose $r=n$. Despite having no explicit rank-regularization, the method converges to a low-rank solution.
>
> **How does rank of X affect qualities**
>
> We get similar results for different choices of rank.

---

> > ### Comment · Reviewer_fryc · 2024-11-26
> >
> > Thank you for the reply.
> >
> > My question was regarding "motivation on why the method is effective for obtaining low rank solution", not asking the authors to summarize certain kinds of phenomenons of the numerical experiments for the $UDV$ decomposition. Also, I emphasize that the $UDV$ decomposition can be covered by a three-layer GLRL except for the constraints, which already encourages low-rank solution. Thus it is unclear why the authors impose these constraints and how/why these constraints can be useful. The whole mechanism is unclear in the current version of this paper.
> >
> > In addition, does *"We get similar results for different choices of rank"* mean that $UDV$ always returns better solution for different rank of $X$?

---

> ### Author Response · Authors · 2024-12-03
>
> We believe our description was clear: as $U$ tends to grow, the projection onto the Frobenius constraint shrinks non-dominant directions. This is why we mentioned the resemblance to the power method and highlighted the role of divergent dynamics in shaping implicit bias, which also clarifies the role of constraints.
>
> We demonstrate that the UDV model naturally exhibits a bias toward truly low-rank solutions without imposing any rank constraints. In contrast, GLRL operates on an entirely different mechanism, manually enforcing low-rankness. Quote from the reference: “At a high level, GLRL is a greedy algorithm that performs rank-constrained optimization and relaxes the rank constraint by 1 whenever it fails to reach a global minimizer of f with the current rank constraint”.
>
> It is unclear to us what the reviewer means by $X$ in this context. We interpret this as referring to the matrix completion problem and the rank of the ground truth matrix. If this interpretation is correct, we obtained low-rank solutions with good recovery accuracies using UDV, for ground truth matrices with different choices of $r$. Otherwise, we would like to emphasize that our approach does not involve a tunable rank parameter, as we use $(n \times n)$ dimensional factors, and the low-rank nature of the solutions comes from implicit bias.

---

> > ### Comment · Reviewer_fryc · 2024-12-03
> >
> > Thank you for the reply.
> >
> > I emphasize that you are considering the paramterization $UDV$, which is certainly a three layer linear neural network that can be covered by Li et al 2020, while the only difference is that the matrices have more constraints. Then the contribution is about the constraints, which, however, is very limited from my perspective. In addition, the author did not provide any compelling theoretical justification on the mechanism of the proposed method to directly answer my question in my original review. Thus I have decided to change my score to 5 to align with other reviewers.

---

### Official Review · Reviewer_6ytx · 2024-10-30

**Soundness:** 1
**Presentation:** 2
**Contribution:** 1
**Rating:** 3
**Confidence:** 5

**Summary:**

In this work the authors introduces a matrix factorization model UDU as an improvement over the traditional BM factorization. The authors claim that while BM factorization find approximate low rank solutions depending on small initialization and lr, the UDU model can find exact low rank solutions independent on initialization scale or stable lr. Then, the authors use this factorization on classification and regression tasks and compare with the two layer UV model.

**Strengths:**

The introduction and the setting of the paper is well established. The proposed method following the lines of past works are clearly written.

**Weaknesses:**

There are several weaknesses in the technical and novelty section of the paper which is highlighted:

*Formulation is not novel enough*


The authors claim that the proposed UDV model leads to faster convergence to low rank solutions irrespective of lr or initialization. However due to lack of analysis, the authors might have missed the fact the model is a mere extension of deep matrix with depth=3 with additional constraints. It is well known that with increasing depth, the solution recovered becomes more low rank. In other words, the residual singular values of the product matrix gets suppresed by the power of depth leading to attenuation of residual singular values, hence stronger low rank. This has been studied in great detail in https://arxiv.org/abs/1905.13655 and followed up in several papers such as https://arxiv.org/abs/2011.13772 and https://arxiv.org/abs/2406.04112. The current model UDV is a depth-3 model with spectral normalization on U and V and diagonal constraint on D. The lack of comparisons or even mention of larger depth models raises question on the novelty of the formulation.

*Unclear contributions*

The authors mention the dynamics is divergent but in their experiments they show no experiments where this is the case. The paper they cite https://arxiv.org/pdf/2005.06398 only shows divergent behaviours for matrix completion and not for matrix sensing problems. Furthermore, the arcthitecture proposed is just a 3 layer network with spectral normalization on the first, last layer and diagonal contraint on the middle one. There seems to be nothing new about this contribution.

*Lack of analysis and insights into the model*

As stated earlier, the slight advatange the UDV model seems to exhibit is most probably due to the effect of depth which promotes increased attenuation of singular values. Even applying SVT (singular value thresholding) post training to higher depth factors seems to give quite good compression rate https://openreview.net/pdf?id=1KCrVMJoJ9. The lack of any analysis for gradient flow on this model prevents from understanding why this model has slight advantage over BM factorization.

**Questions:**

The primary question is how is the proposed model any different from a deep matrix facrtorization setup with depth>3. There are no experiments comparing higher depth. Is there any difference, if so, can the authors compare and provide analysis and insight for gradient flow?

---

> ### Author Response · Authors · 2024-11-25
>
> **Formulation is not novel enough**
>
> The reviewer claims that comparisons with standard depth-3 models are missing, despite these being **explicitly addressed in both the main paper (lines 451–457) and Appendix B5**. Furthermore, we have already cited the references brought up by the reviewer, such as (Arora et al. 2019) and (Feng et al. 2022). We are disappointed with this comment, particularly because the reviewer’s confidence score was 5.
>
> **Unclear contributions**
>
> This paper lies in the observation that the proposed framework consistently converges to truly low-rank solutions. To the best of our knowledge, this behavior has not been reported in prior work and is fundamentally different from achieving approximately low-rank solutions, which are more commonly studied.
>
> We can observe the divergent dynamics especially in matrix factorization experiments. The $UDU^T$ formulation finds factors that are rank-revealing: either the diagonal entries of D or the columns of U converge to zero, leading to truly low-rank solutions. Specifically, U exhibits a tendency to grow along certain directions, while the rescaling induced by projection onto the bounded constraint shrinks other coordinates. This behavior aligns with the mechanisms observed in the power method and provides insights into the connection with divergent forces. We have now included more details on this in the supplementary material.
>
> **Lack of analysis and insights into the model**
>
> As stated earlier, we have already shown in Appendix B5 that standard depth-3 models do not exhibit the same pronounced spectral decay as our proposed framework. In the original submission, this comparison was limited to the HPART dataset for a regression task. To strengthen the evaluation, we have now extended this section. Specifically, we have included additional datasets, present results also using cross-entropy loss in classification tasks, and explore transfer learning scenarios. We posit that constraints provide a tighter and more interpretable mechanism for promoting low-rank behavior, as they explicitly restrict the space for U and V. This advantage is particularly evident in our experiments.
>
> Moreover, prior theoretical work primarily focuses on achieving approximately low-rank solutions, a goal that is theoretically tractable under certain settings and can often be achieved through standard convex optimization formulations, such as those using nuclear norm penalties. In contrast, our empirical results reveal the remarkable observation that the proposed framework consistently converges to truly low-rank solutions. This behavior is fundamentally different and cannot be explained by existing theories on approximate low-rank solutions. While providing theoretical guarantees for convergence to strictly low-rank solutions is beyond the current scope of our work, our findings open intriguing directions for future theoretical research.

---

> > ### Comment · Reviewer_6ytx · 2024-11-26
> > **Reviewer comment**
> >
> > Thanks for the reply. Sorry for the oversight for depth experiments in the appendix. I think this is a key comparison and should have been in the manuscript to highlight the contributions of the paper. Furthermore, the mechanims of rank reduction should be explored in more detail. For example "Specifically, U exhibits a tendency to grow along certain directions, while the rescaling induced by projection onto the bounded constraint shrinks other coordinates."...is there any experiment to validate where this happens? Can you prove this limiting case using the GD iterate updates?
> > Allthough the experiments convey an interesting phenomenon, it is hard to come up with why this happens just by speculation. For these reasons, I maintain my score.

---

### Official Review · Reviewer_QwTg · 2024-10-30

**Soundness:** 2
**Presentation:** 2
**Contribution:** 1
**Rating:** 3
**Confidence:** 4

**Summary:**

The paper looks at the low rank bias of matrix factorization and how to improve upon it. They propose to use a different decomposition of the target matrix and appropriate constraints. The constraints are enforced using a projection. The benefit of the reformulation is shown on a synthetic problem in the paper and additional experiments are provided in the appendix. Moreover, the idea is extended to feedforward neural networks. The paper illustrates using transfer learning that the method is well equipped to prune neurons.

**Strengths:**

The paper is clear in convening the main idea. The idea of the reformulation is to the best of my knowledge novel (although it is directly based on SVD and projected gradient descent). In addition, the extension to neural networks is new. The method outperforms the BM algorithm on several tasks.

**Weaknesses:**

As mentioned by the authors deeper matrix factorization has a lower rank bias. The experiment in appendix B5 shows that just the depth can not explain the bias. This shows the importance of the constraints introduced which act as explicit regularization. Thus, comparing it with explicit regularization such as weight decay is needed. For example, a comparison with weight decay on $U$ and $V$ for the formulation $UV$ which also has been proposed as a pruning strategy in:

https://openreview.net/forum?id=880tEHqxzg

Furthermore, there is no theoretical analysis of the formulation. For neural networks, an unconstrained optimization problem would get worse results on a metric than a constrained one based on that metric. Similarly, which of the constrained formulations leads to the best (convex) domain? In appendix A1 it is seen that the BM algorithm gets lower residual than the proposed method at the cost of a higher rank. This suggests that the constrained loss landscape is excluding good minimizers. Therefore, looking at the loss landscape might be a needed addition.

The neural network experiments are of small scale. To show that the reformulation can be used as a pruning method it has to work for larger scale experiments as well. Note that there are multiple pruning methods to compare with:

https://arxiv.org/abs/2002.03231

https://arxiv.org/abs/2402.19262

**Questions:**

- Could you please compare with weight decay?
- Provide large scale experiments and comparison to pruning benchmarks.
- Motivate the use of particular constraints by theory or loss landscapes.
- Explain the worse performance of the method than the BM algorithm in A1.

---

> ### Author Response · Authors · 2024-11-25
>
> **Comparison against weight decay**
>
> This is a great suggestion. We conducted preliminary experiments using the regression problem with the HPART dataset, comparing the singular value decay observed in depth-2 and depth-3 fully connected neural networks with weight decay regularization. While increasing the regularization parameter led to faster decay in the singular value spectrum, it came at the cost of worse training and validation loss. We tried regularization parameters $10^{-6}, 10^{-5}, 10^{-4}, \dots$, and selected the largest values where the validation loss was at most 10% worse than the UDV baseline (this appeared around $10^{-3}$ or $10^{-2}$, depending on the algorithm). Under these conditions, the resulting singular value decay was not as pronounced as that of the UDV model. When we increased the regularization parameter further to match or surpass the spectral decay of the UDV model, the resulting validation loss became significantly worse than that of the UDV network. These findings highlight that while weight decay regularization does enhance singular value decay, it cannot replicate the results observed with the UDV model.
> We will include this preliminary result in the updated submission and plan to extend this analysis to other settings with classification and transfer learning for the final version.
>
> **Lack of theoretical analysis**
>
> We take this opportunity to clarify the key distinction between achieving approximately low-rank solutions and converging to truly low-rank solutions, which underpins the **novelty, importance, and extraordinary nature of our observations**. Prior theoretical work primarily focuses on achieving approximately low-rank solutions, a goal that is theoretically tractable under certain settings and can often be achieved through standard convex optimization formulations, such as those using nuclear norm penalties. In contrast, our empirical results reveal the remarkable and unexpected observation that the proposed framework consistently converges to **truly low-rank solutions**. This behavior is fundamentally different and cannot be explained by existing theories on approximate low-rank solutions. While providing theoretical guarantees for convergence to strictly low-rank solutions is beyond the current scope of our work, our findings open intriguing directions for future theoretical research.
>
> **Large scale experiments for pruning**
>
> The main contribution of our paper is the extraordinary observation that the UDV model can identify truly low-rank solutions. We believe this observation is of significant interest to researchers studying the theory of neural networks, as the dynamical behavior exhibited by UDV networks provides insights into the underlying mechanisms of implicit bias in fundamentally different ways compared to existing theories based on gradient flow. The pruning experiments in our paper are intended as a proof of concept, demonstrating that truly low-rank solutions can also be leveraged for practical purposes. Our goal in this section is not to provide an exhaustive analysis of pruning techniques but rather to illustrate one potential application of our framework. We acknowledge the extensive literature on pruning via low-rank factorization and note that UDV networks can be seamlessly integrated with existing compression and pruning methods. We adopted singular value thresholding in our experiments only for its simplicity.

---

> > ### Comment · Reviewer_QwTg · 2024-11-26
> >
> > Thank you for your response. I have read it and the comments made by other reviewers. I appreciate the additional weight decay experiment. Although the insight for true low-rank solutions is interesting, my concerns about theory and large scale experiments still remain the same. Additionally, the novelty of the observation appears limited, as noted by several reviewers who referenced prior related works. Therefore, I have decided to maintain my current score.

---

> ### Author Response · Authors · 2024-11-25
>
> **Worse performance of the method than BM in A1**
>
> It is not correct to consider smaller objective value as better performance – given that the goal is finding a low-rank matrix, approximately low-rank matrices provide a relaxation and the smaller objective in training data does not necessarily translate into a good performance in predicting unobserved instances – in other words, the model is overfitting.
>
> To illustrate this, we report the relative distance between the true matrix ($X^{\natural}$) and the matrix recovered $(X)$ using $UU^T$ and $UDU^T$ methods. The relative error is computed as:
> $$ rel.err = ||X^{\natural} - X||_F / || X^{\natural}||_F $$
> and the results are evaluated after $10^6$ iterations (corresponding to the last iterations in Fig. SM1).
>
> | Step Size ($\eta$)   | $UU^\top$   | $UDU^\top$   |
> |-----------------------|-------------|--------------|
> | $10^{-3}$            | 0.4248      | 0.2322       |
> | $10^{-2}$            | 0.6482      | 0.4337       |
> | $10^{-1}$            | 0.6247      | 0.4238       |
>
> | Distance to Origin ($\xi$) | $UU^\top$   | $UDU^\top$   |
> |----------------------------|-------------|--------------|
> | $10^{-4}$                  | 0.7003      | 0.4372       |
> | $10^{-3}$                  | 0.6759      | 0.4374       |
> | $10^{-2}$                  | 0.6482      | 0.4337       |
> | $10^{-1}$                  | 0.6247      | 0.4238       |
> | $10^{0}$                   | 0.6696      | 0.4325       |
>
> These results demonstrate that $UDU^T$ achieves uniformly better generalization than $UU^T$ ⊤ despite the noisy measurements. (Note that we cannot expect a full recovery in this setting, since the measurements are noisy.)
>
> To further evaluate the performance, we repeated the same experiment for the setting of Fig. 1 (noiseless setting). The results are as follows:
>
> | Step Size ($\eta$)   | $UU^\top$         | $UDU^\top$             |
> |-----------------------|-------------------|-------------------------|
> | $10^{-3}$            | 0.0220           | 0.0088                 |
> | $10^{-2}$            | 0.0173           | $2.57 \cdot 10^{-9}$   |
> | $10^{-1}$            | 0.0167           | $2.58 \cdot 10^{-10}$  |
>
> | Distance to Origin ($\xi$) | $UU^\top$         | $UDU^\top$             |
> |----------------------------|-------------------|-------------------------|
> | $10^{-4}$                  | 0.0027           | $2.58 \cdot 10^{-9}$   |
> | $10^{-3}$                  | 0.0060           | $2.57 \cdot 10^{-9}$   |
> | $10^{-2}$                  | 0.0173           | $2.58 \cdot 10^{-9}$   |
> | $10^{-1}$                  | 0.0766           | $2.59 \cdot 10^{-9}$   |
> | $10^{0}$                   | 0.3477           | $2.57 \cdot 10^{-9}$   |
>
> In this noiseless setting, $UDU^\top$ achieves near-perfect recovery, limited only by numerical precision.

---

### Official Review · Reviewer_xaQQ · 2024-10-31

**Soundness:** 2
**Presentation:** 3
**Contribution:** 1
**Rating:** 3
**Confidence:** 4

**Summary:**

The paper introduces the use of a three-factor decomposition X=UDV^T of weight matrices along with a strong weight decay penalty during the training of neural networks. It also includes numerical demonstrations of this method's effectiveness on a small dataset.

**Strengths:**

The paper is generally well-written and pleasant to read. The three-factor decomposition appears to be effective.

**Weaknesses:**

**Limited Novelty:** The reasoning behind the improved compression achieved by three-factor decomposition with weight decay has been previously presented in Proposition 4.2 of [1] and this explanation holds for algorithms including gradient descent. While the original proposition does not enforce a diagonal constraint on D, its proof generalizes to the diagonal case. Could the authors clarify how their approach advances beyond the method proposed in [1]? Notably, [1] utilizes a Frobenius norm penalty rather than a constraint—could the authors discuss the potential benefits of using a Frobenius norm constraint over a penalty if any, considering that the latter might simplify implementation?

**Issues with the Formulation:** The reparameterization approach raises concerns due to its reliance on the constraint ∣∣U∣∣_F≤α, which appears redundant. This constraint can be met by scaling down U and scaling up D, so it does not affect the solution set for the optimization problem. Consequently, optimization problem (4) is not well-defined. If this constraint is intended to influence the solution during the gradient descent update, this section would benefit from revision to avoid presenting (4) with a potentially redundant constraint.

**Lack of Theoretical Insight**: Since the paper focuses on studying the three-factor reparameterization of a single network layer without nonlinear activation, restricted to gradient descent updates, theoretical insights would be highly expected. Prior work in similar settings with two-factor reparameterization has provided theoretical analysis, which suggests that this problem is amenable to theoretical investigation. Given the existing literature, the paper’s empirical results without theoretical support limit its contribution. Could the authors provide some theoretical analysis on aspects such as convergence rates, optimality conditions, or guarantees on the rank of the solution for their proposed method? How does the theoretical behavior of the three-factor decomposition compare to that of two-factor methods in previous work?

**Limited Numerical Experiments:** The experiments conducted in the paper are limited in scope. Could the authors include comparisons with other state-of-the-art compression methods? Additionally, extending the experiments to larger, more challenging datasets like CIFAR-10 or ImageNet would provide a more comprehensive evaluation of the method's effectiveness and scalability.

[1]Zhang, X., Alkhouri, I. R., & Wang, R. (2024). Structure-Preserving Network Compression Via Low-Rank Induced Training Through Linear Layers Composition. arXiv preprint arXiv:2405.03089.

**Questions:**

Please refer to the weaknesses above. I would consider raising the score if these concerns are adequately addressed

---

> ### Author Response · Authors · 2024-11-25
>
> **Lack of Theoretical Insight**
>
> We take this opportunity to clarify the key distinction between achieving approximately low-rank solutions and converging to truly low-rank solutions, which underpins the **novelty, importance, and extraordinary nature of our observations**. Prior theoretical work primarily focuses on achieving approximately low-rank solutions, a goal that is theoretically tractable under certain settings and can often be achieved through standard convex optimization formulations, such as those using nuclear norm penalties. In contrast, our empirical results reveal the remarkable and unexpected observation that the proposed framework consistently converges to **truly low-rank solutions**. This behavior is fundamentally different and cannot be explained by existing theories on approximate low-rank solutions. While providing theoretical guarantees for convergence to strictly low-rank solutions is beyond the current scope of our work, our findings open intriguing directions for future theoretical research.
>
> In matrix factorization, we observe in experiments that the UDU formulation finds factors that are rank-revealing: either the diagonal entries of D or the columns of U converge to zero, leading to truly low-rank solutions. Specifically, U exhibits a tendency to grow along certain directions, while the rescaling induced by projection onto the bounded constraint shrinks other coordinates. This behavior aligns with the mechanisms observed in the power method and provides insights into the connection with divergent forces. We have now included this result in the supplementary material.
>
> **Limited Novelty**
>
> We compared our proposed UDV framework against three-layer fully connected neural networks. In Appendix B5, we provided experimental results demonstrating that our approach exhibits significantly stronger spectral decay than a standard three-layer fully connected network. In the original submission, this comparison was limited to the HPART dataset for a regression task. To address the reviewer's concerns and to strengthen the evaluation, we have now extended this section. Specifically, we have included additional datasets, present results also using cross-entropy loss in classification tasks, and explore transfer learning scenarios. We posit that constraints provide a tighter and more interpretable mechanism for promoting low-rank behavior, as they explicitly restrict the space for U and V. This advantage is particularly evident in our experiments.
>
> **Issues with the Formulation** “This constraint can be met by scaling down U and scaling up D, so it does not affect the solution set for the optimization problem”
>
> Overparameterized neural networks inherently involve redundancy in their parameterization, as this redundancy facilitates learning and often leads to solutions with better generalization properties (e.g., implicit bias toward low-rank solutions in our setting). In nonconvex optimization, algorithms applied to equivalent formulations can (and often do) converge to different solutions due to the inherent sensitivity of the optimization trajectory to the parameterization and constraints. This principle is a cornerstone of neural network training and is precisely why reparameterizations, such as the one we propose, are valuable. In this context, the claim that the model is "not well-defined" fundamentally misunderstands the standard practice of overparameterization in neural network optimization.
>
> **Limited Numerical Experiments:**
>
> We respectfully disagree with the reviewer’s assessment that the experiments are limited in scope. Observing the bias toward truly low-rank solutions requires running the models for extended periods, which is particularly evident in our matrix factorization examples. This constraint naturally limits the scale of datasets we can investigate in the current work. Nonetheless, we have designed our experiments to test this phenomenon across diverse settings, including matrix factorization problem, phase retrieval, neural network training with regression, classification, and transfer learning tasks, with multiple datasets and loss functions. These varied scenarios provide strong evidence of the generality and robustness of our findings. We do, however, plan to explore this framework in the context of self-attention and other architectures in future work.

---

> > ### Comment · Reviewer_xaQQ · 2024-11-27
> >
> > I appreciate the authors' response. However, I would like to reiterate my request for a direct (theoretical and numerical) comparison with [1], given the notable similarities. The method in [1] also employs the Frobenius norm of the factors to penalize the rank and provides exact low-rank solutions with theoretical guarantees when the true matrix is low-rank. The current comparison in Appendix B5 is not exactly with this method.

---

> > > ### Author Response · Authors · 2024-12-03
> > >
> > > Our work fundamentally focuses on constraints rather than regularization, demonstrating that constraints lead to truly low-rank solutions. Please note that we obtained low-rank solutions even in the noisy setting, where the underlying ground-truth matrix is not exactly low-rank. In contrast, the results and theoretical statements in [1] are concerned with approximate low-rank solutions. We will include a discussion on these differences to the related works section. Our main contribution lies in offering a novel perspective on implicit bias through the lens of constrained optimization.

---

> ### Comment · Reviewer_xaQQ · 2024-12-03
>
> Thank you for the reply. The authors mention that the primary advantage of the proposed method is its ability to achieve low-rank reconstruction even in the presence of noise. However, I am not sure why this would be the case. Given that the loss function in equation (4) will ultimately converge to zero, the reconstruction produced by your method would perfectly fit the noisy, high-rank matrix, rather than yielding a low-rank approximation.

---

### Author Response · Authors · 2024-12-04

We appreciate the time and effort the reviewers dedicated to evaluating our work. However, we are disappointed that they failed to recognize, even after the discussions, the fundamental distinction between constraints and regularization. Our paper builds on the observation that Frobenius-norm constraints in matrix factorization (and neural network training) produce truly low-rank solutions. Our proposed formulation produces rank-revealing solutions, where the factors tend to grow in certain directions, while the rescaling induced by projection onto Frobenius norm-ball shrinks other coordinates. This behavior resembles the mechanisms of the power method, drawing connections between the alignment along dominant components and implicit bias. **Our findings are different from the mainstream narrative**, which seems to be one clear reason behind the failure to recognize this paper's main message. We admit, it is our responsibility to clarify the novelty and importance of our results, but the reviewers’ insistence on interpreting our observations within the framework of the mainstream narrative –largely focused on gradient flow dynamics and weight decay regularization– overlooks the crucial **distinction between regularization and constraints on the algorithm's dynamical behavior**, hence misses the core contributions of our work. We argue that **constraints make the implicit bias transparent** and captures the true dynamics. Regularization approximates this behaviour, resulting in approximately low-rank solutions and obscuring the fundamental dynamics that drive implicit bias. We believe **our observations are insightful and hold significant value** for researchers exploring implicit bias.

---

### Meta-Review · Area_Chair_h1c2 · 2024-12-18

**Metareview:**

In this work the authors propose a reparameterization of matrix factorization and neural network linear layer operators into the product of three matrices, where the middle matrix is constrained to be non-negative and diagonal and the left and right matrices are constrained to have bounded Frobenius norm.  The authors argue that the motivation for doing so is to achieve a more pronounced implicit bias towards a low-rank solution due to the optimization dynamics that result from the reparameterization.  The reviewers are all somewhat in agreement that the paper is not ready for publication as the proposed model is somewhat incremental to other works, the paper lack any theoretical analysis, and there are missing comparisons to other closely related works/benchmarks.

Overall, while I find the idea intriguing, I am in agreement with the reviewers that the paper is not yet meeting the bar for publication at ICLR.  There are numerous potential reparameterizations and optimization algorithms one could propose that might be expected to encourage low-rank solutions based on optimization dynamics, but one of the core contributions of the implicit bias literature is to show theoretically that (under certain conditions, loss functions, etc) the optimization dynamics will provably converge to these low-rank solutions.  Unfortunately, much of the argument here appears to be largely empirical, which, while perhaps encouraging, does not meet the typical standard for papers on this topic.

One point highlighted by the authors is that the dynamics induced by their formulation result in "exact" low-rank solutions.  While this is perhaps desirable, it is not particularly surprising to me, since as soon as one of the diagonal elements goes negative during gradient descent the projection onto the non-negative orthant (presumably via simply clipping negative entries) lowers the rank of the factorization.  However, this is a double-edged sword, as depending on the choice of initialization, step-size, etc one could easily truncate the rank of the factorization prematurely from which the algorithm can never recover.  I would encourage the authors to consider theoretical analysis to establish under what conditions the formulation can be guaranteed to succeed.

**Additional Comments On Reviewer Discussion:**

The authors have provided a rebuttal to many of the points raised by the reviewers, but after reading the rebuttals many of the reviewers still have concerns regarding the paper and do not recommend acceptance.

---

### Decision · Program_Chairs · 2025-01-22

Reject